

# Musculoskeletal modelling of an ostrich (*Struthio camelus*) pelvic limb: influence of limb orientation on muscular capacity during locomotion

John R. Hutchinson[1,2], Jeffery W. Rankin[1], Jonas Rubenson[3,4], Kate H. Rosenbluth[2], Robert A. Siston[2,5] and Scott L. Delp[2]

[1] Structure and Motion Laboratory, Department of Comparative Biomedical Sciences, The Royal Veterinary College, University of London, Hatfield, Hertfordshire, United Kingdom
[2] Bioengineering Department, Stanford University, Stanford, CA, USA
[3] School of Sport Science, Exercise and Health, The University of Western Australia, Perth, WA, Australia
[4] Department of Kinesiology, The Pennsylvania State University, University Park, PA, USA
[5] Department of Mechanical and Aerospace Engineering, The Ohio State University, Columbus, OH, USA

Corresponding author
John R. Hutchinson,
jhutchinson@rvc.ac.uk

## ABSTRACT

We developed a three-dimensional, biomechanical computer model of the 36 major pelvic limb muscle groups in an ostrich (*Struthio camelus*) to investigate muscle function in this, the largest of extant birds and model organism for many studies of locomotor mechanics, body size, anatomy and evolution. Combined with experimental data, we use this model to test two main hypotheses. We first query whether ostriches use limb orientations (joint angles) that optimize the moment-generating capacities of their muscles during walking or running. Next, we test whether ostriches use limb orientations at mid-stance that keep their extensor muscles near maximal, and flexor muscles near minimal, moment arms. Our two hypotheses relate to the control priorities that a large bipedal animal might evolve under biomechanical constraints to achieve more effective static weight support. We find that ostriches do not use limb orientations to optimize the moment-generating capacities or moment arms of their muscles. We infer that dynamic properties of muscles or tendons might be better candidates for locomotor optimization. Regardless, general principles explaining why species choose particular joint orientations during locomotion are lacking, raising the question of whether such general principles exist or if clades evolve different patterns (e.g., weighting of muscle force–length or force–velocity properties in selecting postures). This leaves theoretical studies of muscle moment arms estimated for extinct animals at an impasse until studies of extant taxa answer these questions. Finally, we compare our model's results against those of two prior studies of ostrich limb muscle moment arms, finding general agreement for many muscles. Some flexor and extensor muscles exhibit self-stabilization patterns (posture-dependent switches between flexor/extensor action) that ostriches may use to coordinate their locomotion. However, some conspicuous areas of disagreement in our results illustrate some cautionary principles. Importantly, tendon-travel empirical measurements of muscle moment arms must be carefully designed to preserve 3D muscle geometry lest

their accuracy suffer relative to that of anatomically realistic models. The dearth of accurate experimental measurements of 3D moment arms of muscles in birds leaves uncertainty regarding the relative accuracy of different modelling or experimental datasets such as in ostriches. Our model, however, provides a comprehensive set of 3D estimates of muscle actions in ostriches for the first time, emphasizing that avian limb mechanics are highly three-dimensional and complex, and how no muscles act purely in the sagittal plane. A comparative synthesis of experiments and models such as ours could provide powerful synthesis into how anatomy, mechanics and control interact during locomotion and how these interactions evolve. Such a framework could remove obstacles impeding the analysis of muscle function in extinct taxa.

## INTRODUCTION

As the largest living avian bipeds, ostriches (*Struthio camelus* Linnaeus 1758) are important for understanding how body mass influences locomotor mechanics in birds. In addition, ostriches are among the fastest of living terrestrial animals, and are the fastest living (perhaps even the fastest ever) bipedal runners. These birds can reach maximum speeds $>15$ ms$^{-1}$ (*Alexander et al., 1979*); similar to another biped that is coincidentally of similar size: red kangaroos (*Macropus rufus*) (*Bennett & Taylor, 1995*). Examination of their locomotor dynamics may reveal some of the complex factors that determine maximum running speed in land animals and guide the development of fast running machines. Ostriches are also of similar body size to humans, which other than birds are the only obligate striding bipeds today, making comparisons of bipedal locomotor function in these two species possible (e.g., *Gatesy & Biewener, 1991*; *Rubenson et al., 2011*). Additionally, as the largest extant birds, ostriches are important "endpoints" for studies of body size effects on locomotion (e.g., *Maloiy et al., 1979*; *Gatesy, Bäker & Hutchinson, 2009*; *Brassey et al., 2013a*; *Brassey et al., 2013b*; *Kilbourne, 2013*). Furthermore, ostriches are members of the ratite bird clade, whose evolution from basal flying birds into large cursorial flightless animals has been of longstanding scientific interest. However, the evolutionary patterns and processes that produced the diversity of living ratites and their unusual locomotor mechanisms remain uncertain (*Baker et al., in press* and references therein). In turn, ratite birds including ostriches occupy relatively basal positions in extant avian phylogeny (e.g., *Cracraft & Clarke, 2001*; *Baker et al., in press*). Despite their remarkable cursorial specializations and evolutionarily increased body size, ostriches can offer clues to the structure and function of earliest crown clade birds, and thereby about the evolution of avian locomotion from their theropod dinosaur forebears (e.g., *Gatesy, 1990*). Ostriches have also often been employed as analogues for dinosaur locomotion (e.g., *Russell, 1972*; *Paul, 1998*) despite some major anatomical differences (*Gatesy, 1990*; *Gatesy, 1995*; *Hutchinson & Gatesy, 2000*; *Hutchinson,*

*2002*; *Gatesy, Bäker & Hutchinson, 2009*; *Hutchinson & Allen, 2009*), so ostriches and extinct dinosaurs may be reciprocally informative.

A general problem facing those interested in examining the above questions in detail is that ostrich locomotion, including pelvic limb structure-function relationships during movement, remains incompletely understood. Numerous studies have empirically investigated the locomotor kinematics and kinetics of ostriches (e.g., *Alexander et al., 1979*; *Gatesy & Biewener, 1991*; *Abourachid, 2001*; *Abourachid & Renous, 2000*; *Rubenson et al., 2004*; *Rubenson et al., 2007*; *Rubenson et al., 2011*; *Smith et al., 2006*; *Smith et al., 2007*; *Smith, Jespers & Wilson, 2010*; *Smith & Wilson, 2013*; *Jindrich et al., 2007*; *Schaller et al., 2009*; *Schaller et al., 2011*) and much focus has been given to the whole-body mechanics and energetics of ostriches (e.g., *Fedak & Seeherman, 1979*; *Fedak, Heglund & Taylor, 1982*; *Rubenson et al., 2004*; *Watson et al., 2011*). However, detailed understanding of the interactions between the various components of the musculoskeletal system and the environment during these movements remain poorly understood. Due to the complex, non-linear dynamics of the musculoskeletal system, correlating whole-body level measures of locomotion to specific muscle function-structure relationships cannot yet be adequately performed.

Furthermore, ostrich myology was not carefully described until recently (*Weissengruber, Forstenpointner & Gangl, 2003*; *Gangletal, 2004*; *Zinoviev, 2006*; also knee joint functional morphology by *Fuss, 1996*; foot/ankle function by *Schaller et al., 2009*, *Schaller et al., 2011*). Previous myological studies were marred by errors in interpreting ostrich anatomy and by confusing application of mammalian anatomy to ostriches (e.g., *Haughton, 1864*; *Macalister, 1864*) or provided only a superficial treatment of proximal pelvic limb anatomy (e.g., *Mellett, 1994*). Even *Gadow (1880)*, a classic comparative reference, contains antiquated concepts of homology (*Rowe, 1986*). These prior studies of ostrich myology augment comparative work on the pelvic limb myology of other ratites (e.g., *Haughton, 1867a*; *Haughton, 1867b*; *Gadow, 1880*; *McGowan, 1979*; *Vanden Berge, 1982*; *Patak & Baldwin, 1998*; *Picasso, 2010*; *Picasso, 2012*; *Chadwick et al., 2014*; *Lamas, Main & Hutchinson, 2014*; *Regnault, Pitsillides & Hutchinson, 2014*). Additionally, data are available on muscle physiology in ostriches (*Velotto & Crasto, 2004*) and other ratites (e.g., *McGowan, 1979*; *Patak & Baldwin, 1993*), although biomechanical data characterizing muscle force–velocity and force–length relationships for avian pelvic limb muscles are scant (e.g., *Nelson, Gabaldón & Roberts, 2004*).

This body of prior research provides the strong foundation necessary for detailed examination of ostrich limb muscle function using anatomically-realistic biomechanical modelling in order to advance understanding of how the largest living bird supports its body weight and moves itself with its pelvic limb muscles. Here, to provide new insight into ostrich locomotor structure-function relationships, we investigate how pelvic limb muscle functions relate to limb orientation (i.e., posture/pose or joint angles; *Gatesy, 1995*; *Gatesy, Bäker & Hutchinson, 2009*) in ostriches. To do this, we integrate data from experimentally measured joint kinematics and ground reaction forces with a biomechanical computer model that was constructed by digitizing the bones, muscles, and tendons of an adult

ostrich. By replicating ostrich structure and behaviour, the musculoskeletal model provides estimates of individual muscle moment arms (*Pandy, 1999*) and maximum capacity for moment generation during those behaviours: quantities that are otherwise difficult or impossible to accurately measure non-invasively, especially for all thirty-six major pelvic limb muscles simultaneously. Here we use this model to address some fundamental mysteries about locomotion in ostriches as well as birds, non-avian dinosaurs and bipeds.

Larger mammalian species tend to have straighter limbs than smaller species to improve their effective mechanical advantage during movement (*Biewener, 1989*; *Biewener, 1990*). Yet it remains unclear how much of this improvement is achieved by reducing the moment arms of ground reaction forces about the limb joints using less flexed limb joint orientations or by increasing muscle moment arms via increased anatomical leverage (e.g., relatively larger trochanters) or straightened limb orientation (i.e., which shifts muscles further away from joint centres; *Hutchinson et al., 2005*). Our ostrich musculoskeletal model will facilitate discerning this relationship in birds, for whom it seems a similar pattern to mammals of having improved mechanical advantage in larger species holds (*Gatesy & Biewener, 1991*; *Hutchinson, 2004*; *Günther et al., 2004*; *Brassey et al., 2013a*; *Kilbourne, 2013*).

Previous studies (e.g., *Hutchinson et al., 2005*; and references therein) suggested that limb antigravity muscle moment arms (or moment-generating capacity; *Full & Ahn, 1995*) may peak in very upright limb orientations, which intimates that smaller animals with more crouched poses (e.g., birds) employ sub-optimal joint angles for supporting their body weight (*Biewener, 1989*; *Günther et al., 2004*), presumably as a tradeoff to provide other benefits such as increased manoeuvrability (e.g., *Daley & Usherwood, 2010*). As the largest living birds, ostriches stand and move with straighter limbs than smaller birds (*Gatesy & Biewener, 1991*). However, ostriches are bipeds that still habitually support themselves with markedly flexed hip and knee joints, which make them a useful case study of this mechanical relationship between posture and antigravity muscle capacity. Understanding this relationship impacts the broader question of why animals choose certain postures—do they select postures that favour larger moment arms for economical force production (e.g., *Fujiwara, 2009*; *Fujiwara, Endo & Hutchinson, 2011*; *Fujiwara & Hutchinson, 2012*) or other factors such as muscle force–length properties (e.g., *McClearn, 1985*; *Lieber & Boakes, 1988a*; *Lieber & Boakes, 1988b*; *Lieber & Brown, 1992*; *Lieber & Shoemaker, 1992*; *Lieber, 1997*)?

The moment arms of ostrich pelvic limb muscles have been studied before with two different methodologies: two-dimensional (2D) experimental ("tendon travel") measurements of de-fleshed limbs (*Smith et al., 2007*) and 3D musculoskeletal computer models constructed from literature data, dissections and scanned skeletons (*Bates & Schachner, 2012*). The question remains open: how accurate are these experimental measurements and models, and how consistently can different researchers construct such models given the inherent subjectivity involved? This methodological question impacts many of the questions above; a weak model impairs the ability to test hypotheses. Here, we use our 3D musculoskeletal model and previously collected experimental data to

address three main questions related to the issues described above: (1) Do ostriches adopt limb orientations during walking or running that optimize their capacity to generate maximal moments about the pelvic limb joints? (2) Are the moment arms of limb muscles maximized (for antigravity/extensor muscles; or minimized in the case of antagonistic flexor muscles) at mid-stance of locomotion (optimizing weight support), or at highly extended limb orientations, as prior studies of dinosaurs inferred (e.g., *Hutchinson et al., 2005*)? Finally, (3) how accurate or repeatable are estimates of limb muscle moment arms in ostriches using different methods?

We also integrate our results with previous studies of ostriches and other large birds (cited above) to infer how the pelvic limb muscles function in locomotion. This provides a three-dimensional perspective on avian musculoskeletal function, a necessary shift away from past planar (2D) simplifications of this system (e.g., *Alexander et al., 1979*; *Abourachid, 2001*; *Hutchinson, 2004*; *Gatesy, Bäker & Hutchinson, 2009*) because recent studies have shown complex 3D limb dynamics during avian movement (*Gatesy, 1994*; *Hutchinson & Gatesy, 2000*; *Rubenson et al., 2007*; *Rubenson et al., 2011*; *Goetz et al., 2008*; *Abourachid et al., 2011*; *Andrada et al., 2013*; *Kambic, Roberts & Gatesy, 2014*). Finally, we synthesize our results with similar data from other bipeds, including humans and *Tyrannosaurus rex*, to infer how limb muscle moment-generating capacity more generally relates to limb orientation and body size.

## MATERIALS AND METHODS

### Animals

Experimental biomechanics data were collected from three female adult ostriches (70.0, 78.7, 75.9 kg body mass), from which a single representative animal's data was used as model input (78.7 kg). Birds were housed in a large outdoor paddock (5,000 m$^2$) and provided with unlimited access to food and water. All experiments were performed in accordance with the Animal Ethics Committee of the University of Western Australia. Architectural and geometric data for the muscle–tendon units, and skeletal dimensions used in the computer model, were measured in a third female ostrich (65.3 kg), which had no musculoskeletal pathologies and was culled from a commercial ostrich herd (The Ostrich Meat Company, Merced, California). We subsequently verified these anatomical data by qualitative comparisons with three other adult specimens of qualitatively similar size, as well as literature descriptions (*Gangl et al., 2004*; *Zinoviev, 2006*).

### Kinematic measurements

The experiments and 3D kinematic analyses are described in detail elsewhere (*Rubenson et al., 2004*; *Rubenson et al., 2007*; *Rubenson et al., 2011*). Briefly, the birds were trained to walk and run across a 50 m long fenced runway surfaced with high-density rubber matting (10 mm thickness). An eleven-parameter direct linear transformation (DLT) was used to construct a 3D image volume from two high-speed cameras (200 Hz) positioned at 45° angles to the runway (Peak Motus; Peak Performance, Centennial, Colorado, USA). The DLT was calculated using a custom-built 48-point moveable calibration frame. The

 

video footage from the two cameras was gen-locked and synchronized manually using a hand held switch that caused a barcode in each video field to turn white. All data collection was performed after sunset under artificial lighting. A wide range of speeds was recorded in prior studies (*Rubenson et al., 2004*; *Rubenson et al., 2007*; *Rubenson et al., 2011*), but we only used representative (i.e., 1 trial/gait) kinematic data for a single walking trial at 1.22 ms$^{-1}$ and slow running (at 3.46 ms$^{-1}$). These two trials were within 1 S.D. of the mean kinematic data for running trials from *Rubenson et al. (2007)* and previously unpublished walking data from the same experimental analysis (see 'Results').

The 3D position and orientation of the ostrich limb segments were determined by videotaping clusters of non-linear, retro-reflective markers placed on the pelvis, femur, tibiotarsus, and tarsometatarsus and a single marker placed on the end of the third phalanx. These markers were used to define segment rigid-body Technical Coordinate Systems (TCS; *Cappozzo et al., 1995*). Prior to walking and running trials, static calibration trials were performed in order to identify several key anatomical landmarks necessary to construct segment Anatomical Coordinate Systems (ACS; see *Rubenson et al. (2007)* and *Rubenson et al. (2011)* for details). Anatomical landmarks were identified using a 6-marker pointer device and expressed in the segment TCSs. The static calibration trials thus allowed for the reconstruction of each segment ACS across walking/running strides using the motion data of the segment marker clusters alone.

Three-dimensional marker trajectory data were filtered using a fourth order zero-lag Butterworth low-pass filter (4–12 Hz) and compiled in c3d format (Motion Lab Systems, Baton Rouge, Louisiana, USA). Kinematics were computed using BodyBuilder modelling software (Oxford Metrics, Oxford, UK). Joint angles were calculated by determining the Euler angles associated with the transformation between the ACSs of the proximal and distal segments of a joint (*Grood & Suntay, 1983*). Segment ACSs in both the model (below) and in the experimental animals were constructed using the same landmarks (anatomical landmarks and numerically derived axes) and same ordered set of rotations between proximal and distal ACSs. Thus, 3D joint motion was defined equivalently in the musculoskeletal model and in the experimental animals and we were able to use experimentally derived joint angles as input into our musculoskeletal model to estimate muscular mechanics *in vivo* during walking and running.

### 3D coordinate systems for anatomical dissection

We used a Polaris optical tracking system (Northern Digital Inc., Waterloo, Ontario) to record the positions of anatomic landmarks and relevant joint kinematics during dissection sessions. This tracking system is accurate to within 1.5 mm with the 1.5 m$^3$ measurement volume used in this study (Traxtal Inc., Toronto, Ontario), and in order to ensure this accuracy we performed appropriate calibrations before collecting our data.

We first skinned the right pelvic limb of the ostrich specimen. Before dissection of the muscles, we attached LED-emitting reference frames (AdapTrax trackers; Traxtal Inc., Toronto, Ontario) to each bone segment using orthopaedic bone screws. Each reference frame contained a cluster of LEDs that allowed the tracking system to record

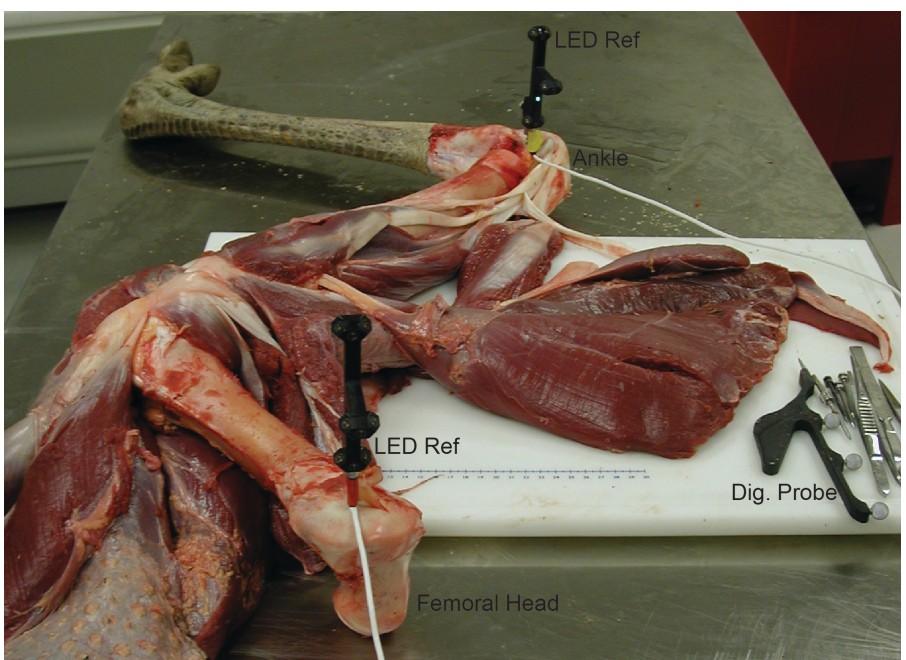

**Figure 1 Digitizing apparatus used during anatomical dissection of ostrich.** "LED Ref" indicates the proximal (in trochanteric crest of the femur) and distal (in tibiotarsus by the ankle) reference frames, "Dig. Probe" indicates the digitizing probe used to collect landmarks.

the 3D position and orientation of each segment (establishing the segment TCSs for the dissections, comparable to that for the experiments). Figure 1 shows the apparatus we used. We used a digitizing probe (Northern Digital Inc., Waterloo, Ontario) to digitize the 3D coordinates of the musculoskeletal geometry in each session relative to these trackers. Unlike the LED-emitting reference frames, the digitizing probe had a cluster of highly reflective spheres, making it an untethered and mobile tool. When these spheres were visible to the tracking system, the 3D position of the tip of the probe (calibrated in advance) could be recorded with respect to the TCS. Three rigid permanent points (marked with a drill as points on the bones) were measured on each segment to provide a local bone coordinate system for all digitizing/dissection sessions. This step allowed the TCS to be removed from the bone and reattached in a different area to facilitate the dissection process while still preserving the overall relationship of digitized points on a given bone between sessions.

Building a musculoskeletal model required points to be expressed in the segment ACSs (Fig. 2 and *Rubenson et al., 2007*; *Rubenson et al., 2011*). The pelvis reference frame was defined as follows: the origin at the midline of the pelvis halfway between the left and right side hip joint centres; the unit vector SUL SYN (*x*-axis; positive being cranial); the cross-product of the unit vector SUL IL and the *x*-axis (*y*-axis; positive being dorsal), and cross-product of the *x*-axis and *y*-axis (*z*-axis; positive being to the right). To locate the hip joint centres, we digitized 10–20 points in and around the acetabulum and femoral head, and then used least-squares optimization to fit a sphere to each of the two resulting point

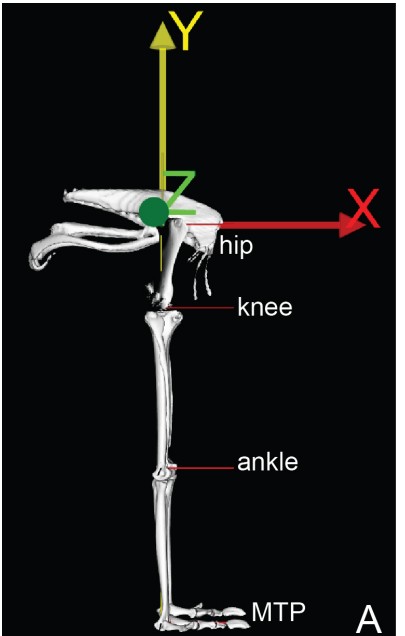
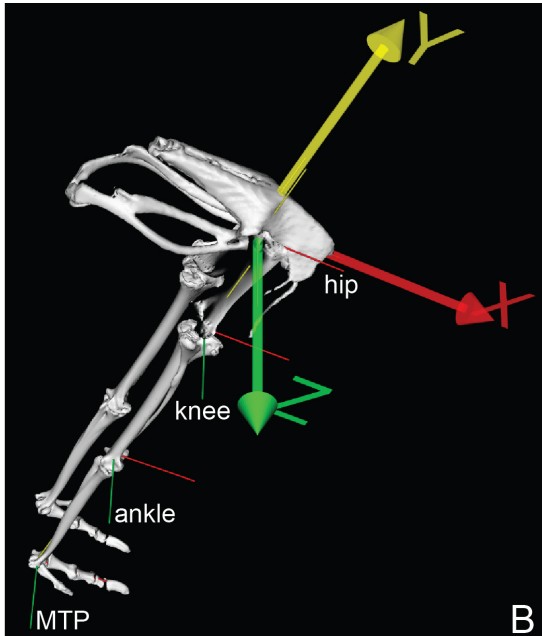

**Figure 2** Ostrich model joint axes $(x, y, z)$ shown in right lateral (A) and oblique right dorsolateral (B) views. The $x$-axis corresponded to ab/adduction, the $y$-axis to long-axis rotation, and the $z$-axis to flexion/extension.

clouds. The centre of this best-fit sphere was the hip joint centre. To establish the reference frames for the other segments, we first estimated the medial-lateral joint rotational axis for the remaining joints by flexing and extending each joint and recording the 3D position and orientation of the distal bone with respect to the proximal one as a series of homogeneous transformation matrices. With these transformation matrices, we were able to calculate the average kinematic screw (helical) axes (*Bottema & Roth, 1990*) that best approximated the flexion-extension axis between those segments.

The femur coordinate system was defined as: the origin at the proximal joint centre; the segment $z$-axis along the medial-lateral joint rotational axis (positive being lateral); the $y$-axis as the cross-product of the $z$-axis and the unit vector between the proximal and distal joint centres; and the $x$-axis as the cross-product of the $y$- and $x$-axes. The tibiotarsus and tarsometatarsus coordinate systems were defined as: the origin at the proximal joint centre; the $y$-axis as unit vector between the proximal and distal joint centres; the segment $z$-axis as the cross product of the medial-lateral joint rotational axis and the $y$-axis; and the $x$-axis as the cross-product of the $y$- and $z$-axes. The pes coordinate system was defined as: the origin at the proximal joint centre; the segment $x$-axis as the unit vector between the proximal joint centre and the end of the segment; the $z$-axis as the cross product of the medial-lateral joint rotational axis and the $x$-axis; and the $y$-axis as the cross-product of the $x$- and $z$-axes. Putting any digitized points into these ACSs required two linear transformations: from the TCS into the local bone coordinate system and subsequently into the ACS. Table 1 provides data on axis positions used in the final model.

**Table 1 Joint axes for the ostrich musculoskeletal model.** Each joint centre is listed in $(x, y, z)$-coordinate space as a distance from the segment origin. The pes was 0.141 m long and an interphalangeal joint's location is noted here in the final row, but was not included in the model. Each joint was defined relative to the one proximal to it, with the pelvis segment placed at the origin of the world coordinate system.

| Joint or segment | Centre $x$ (m) | Centre $y$ (m) | Centre $z$ (m) | Motion axes | Ranges of motion (°) |
|---|---|---|---|---|---|
| Pelvis | 0 | 0 | 0 | $x, y, z$ | [−180/180; −180/180; −180/180] |
| Hip (acetabular/antitrochanteric) | 0 | 0 | 0.0355 | $x, y, z$ | [−45/45; −45/45; −65/10] |
| Knee (femorotibial) | 0 | −0.2338 | 0.0543 | $x, y, z$ | [−45/45; −45/45; −180/10] |
| Ankle (intertarsal) | 0 | −0.442 | 0 | $x, z$ | [−45/45; −10; −10/180] |
| Metatarsophalangeal (MTP III) | 0 | −0.426 | 0 | $z$ | [5; 24; −180/90] |
| [Proximal interphalangeal (D III)] | [0 | −0.089 | 0] | [$z$] | [not estimated] |

## Anatomical digitization and musculoskeletal model construction

We began by dissecting the specimen proceeding from superficial to deep structures. The positions of muscles and bone geometry or other relevant anatomical features on each limb segment were measured using the digitizing probe. In particular, before removing muscles we digitized the circumferences of muscle origins and insertions as well as the 3D paths of the muscles from origin to insertion, using from 1 to 30 (depending on extent of the structure) $x$, $y$, $z$ coordinate points to characterize each structure of interest. In addition, we measured other musculoskeletal features used as references, such as bone surfaces (for later alignment of complete 3D bone images from CT data), condylar contours, and those ligaments that influence muscle–tendon unit paths. All 3D points for use in the model were converted into the ACS.

After dissection the bones were de-fleshed and macerated, but with articular cartilages and menisci remaining as intact as possible. The bones were then CT scanned (1.5–3 mm slices, 120–130 kPa, 109–150 mA, on a Picker PQ5000 CT scanner) and the resulting slices digitized in Mimics software (Materialise, Inc., Leuven, Belgium) to produce rendered 3D polygonal meshes. Ultimately ASCII (ASC) format images were used (decimated to <50,000 polygons each) as bone images in the model. Importantly, the bones were only visual aids and not inherent obstacles to joint or muscle motion, but they are crucial for visualizing musculoskeletal function.

We used these anatomical data and bone images to construct a 3D musculoskeletal computer model of the right pelvic limb using SIMM software (Musculographics, Inc., Chicago, Illinois, USA; *Delp et al., 1990*; *Delp & Zajac, 1992*; *Delp & Loan, 1995*; *Delp & Loan, 2000*). The right limb was mirrored as a left limb. The digitized muscle paths were used as a template for developing the muscle paths in the final model (Figs. 3 and 4). Paths were represented using a combination of "via points" (i.e., static points fixed relative to a segment) and "wrapping surfaces" that prevented translation of points outside of a predefined area (see *Delp & Loan, 1995*; *Delp & Loan, 2000* for details; also *Hutchinson et al., 2005*, for a similar procedure we used for *Tyrannosaurus rex*). Table 2 presents the muscles modelled, with abbreviations used throughout the paper. Table 3 describes all the muscle wrapping surfaces assumed in the model and Fig. 5 displays examples.

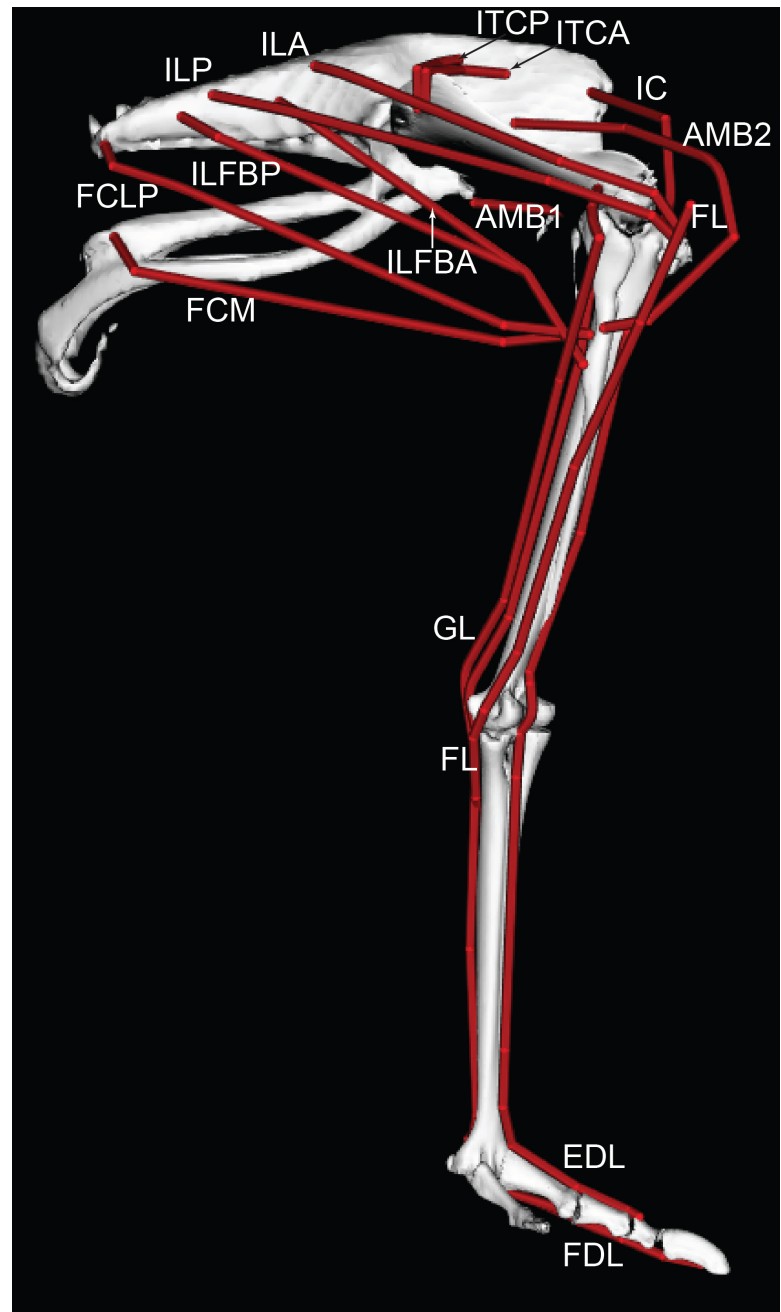

**Figure 3 Ostrich musculoskeletal model in right lateral view, with muscle–tendon units labelled (red lines).** See Table 2 for muscle abbreviations.

Inevitably, because the 3D musculoskeletal geometry was complex, we had to cautiously judge where to position wrapping surfaces and what size and shape they should be. Additional ostrich cadaveric material was used along with the literature (*Gangletal, 2004*; *Zinoviev, 2006*) to qualitatively refine the model as we iteratively progressed, checking that paths and attachments were represented reasonably and consistently. We took care to visualize the model in many different 3D joint positions to ensure that muscles did

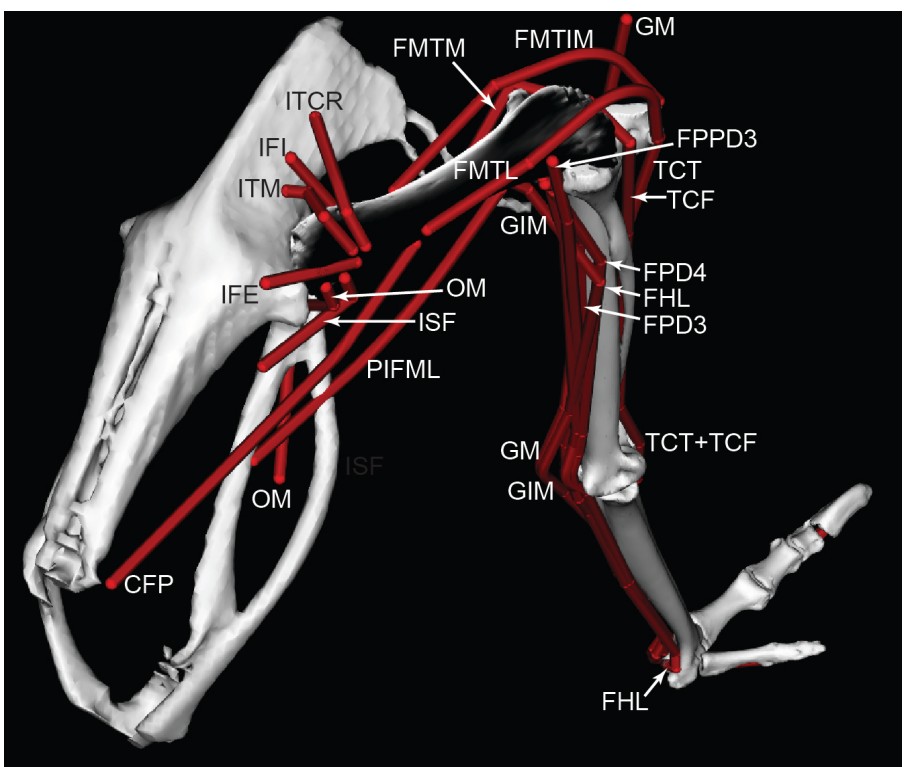

**Figure 4 Ostrich musculoskeletal model in right caudolateral view, with muscle–tendon units labelled (red lines).** See Table 2 for muscle abbreviations.

not pass through areas occupied by other soft tissues or especially bones and to eliminate other numerical errors generated by interactions of the muscle–tendon unit paths with wrapping surfaces (e.g., "loops" in muscles caused by contradictory constraints in the model). Importantly, because we intended to compare our model's results with data from *Smith et al. (2007)* and *Bates & Schachner (2012)*, we kept our model construction blind to the results of these studies, avoiding any comparisons and indeed finishing the major steps in completing our model before these studies were published.

## Muscle–tendon unit architecture and physiology

After we dissected, digitized, and removed the muscles, we separated them from their proximal/distal tendons and other connective tissue. We then used digital calipers ($\pm0.1$ mm), an electronic balance ($\pm0.001$ g), and a protractor ($\pm1°$) to measure muscle fascicle lengths ($L$), masses ($m_{\text{musc}}$), and resting pennation angles ($\theta$) for calculating physiological cross-sectional area ($A_{\text{phys}}$), taking an average of five randomized measurements for $L$ and $\theta$ in larger muscles.

Using water displacement (immersing sectioned muscles in graduated cylinders) to calculate muscle belly (sans tendon) density ($d$) from (volume $m_{\text{musc}}^{-1}$), we obtained a mean value of $1.0645 \times 10^3$ ($n = 10$; S.D. $= 0.0347$) kg m$^{-3}$, matching measurements of mammalian muscle (*Mendez & Keys, 1960*; *Brown et al., 2003a*). Hence, we assumed a conventional value of $d$ as $1.06 \times 10^3$ kg m$^{-3}$. As commonly practiced, we assumed $L$ to

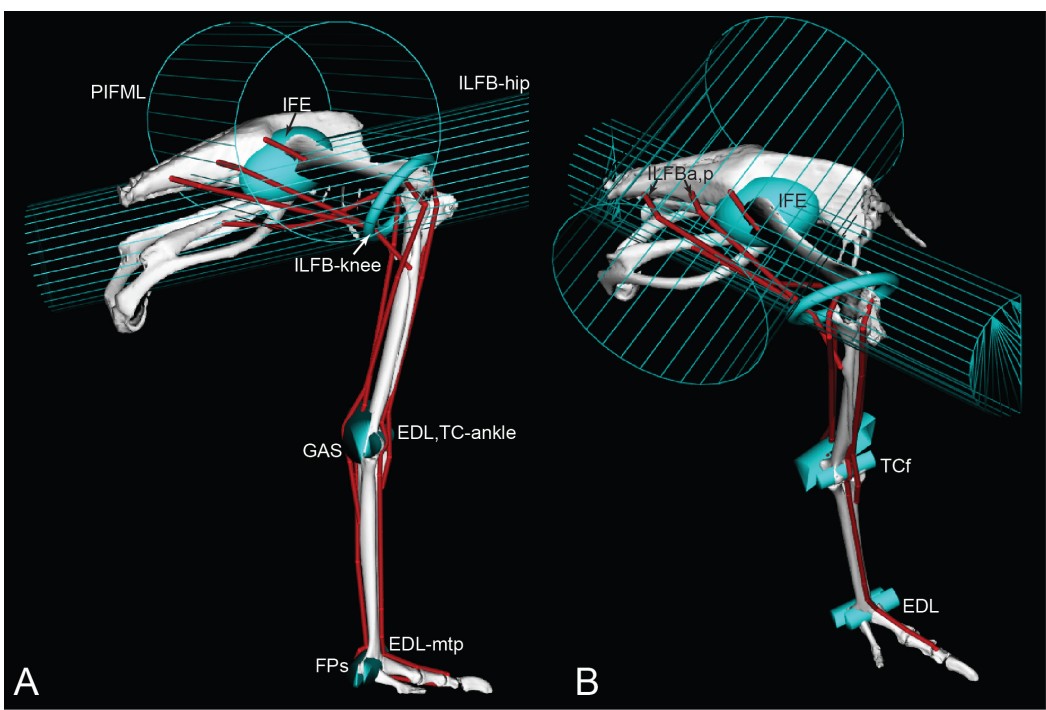
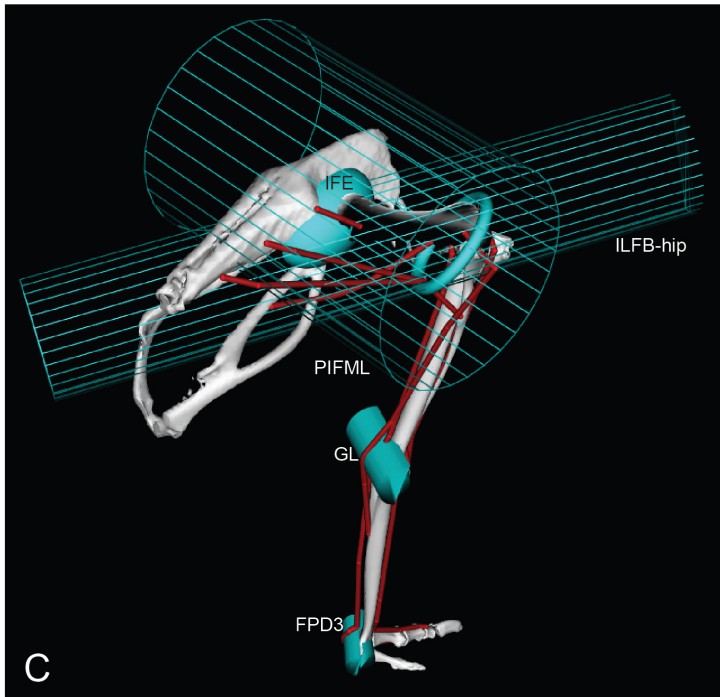

**Figure 5 Ostrich musculoskeletal model: wrapping surface examples.** See Table 2 for muscle abbreviations. Lateral (A), craniolateral (B), and caudolateral (C) views of eight muscle wrapping objects (in blue), as half and whole cylinders, ellipses and a torus. The PIFML and ILFB wrapping surfaces are shown as meshes, for added clarity.

**Table 2 Muscles included in the ostrich musculoskeletal model (ordered as anatomical/functional groups as per prior studies), with their associated abbreviations and physiological/architectural parameters.** Data were obtained via dissection. Blank cells for muscle masses (ILp, ILFBp, ITCp, TCt) indicate that the second part of the muscle shares the mass value, which was divided equally to calculate $A_{phys}$ and hence $F_{max}$.

| Muscle abbreviation | Muscle full name | Muscle mass; $m_{musc}$ (kg) | Fascicle length; $L$ (m) | Pennation angle; $\theta$ (°) | Maximal isometric force; $F_{max}$ (N) |
|---|---|---|---|---|---|
| IC | M. iliotibialis cranialis | 0.3788 | 0.174 | 0 | 615 |
| ILa | M. iliotibialis lateralis (cranial part) | 1.074 | 0.174 | 0 | 875 |
| ILp | M. iliotibialis lateralis (caudal part) | | 0.174 | 0 | 875 |
| AMB1 | M. ambiens, ventral (pubic) head | 0.093 | 0.039 | 10 | 672 |
| AMB2 | M. ambiens, dorsal (iliac) head | 0.1994 | 0.044 | 15 | 1,240 |
| FMTL | M. femorotibialis lateralis | 0.3181 | 0.088 | 15 | 992 |
| FMTIM | M. femorotibialis intermedius | 0.387 | 0.084 | 25 | 1,180 |
| FMTM | M. femorotibialis medialis | 0.272 | 0.089 | 30 | 753 |
| ILFBa | M. iliofibularis (cranial part) | 1.0623 | 0.176 | 0 | 867 |
| ILFBp | M. iliofibularis (caudal part) | | 0.176 | 0 | 867 |
| ITCa | M. iliotrochantericus caudalis (cranial part) | 0.3114 | 0.064 | 25 | 622 |
| ITCp | M. iliotrochantericus caudalis (caudal part) | | 0.064 | 25 | 622 |
| IFE | M. iliofemoralis externus | 0.03264 | 0.025 | 25 | 331 |
| ITM | M. iliotrochantericus medius | 0.0256 | 0.058 | 0 | 125 |
| ITCR | M. iliotrochantericus cranialis | 0.0432 | 0.053 | 10 | 228 |
| IFI | M. iliofemoralis internus | 0.0407 | 0.041 | 0 | 284 |
| FCM | M. flexor cruris medialis | 0.1192 | 0.036 | 35 | 767 |
| FCLP | M. flexor cruris lateralis pars pelvica | 0.3182 | 0.24 | 0 | 376 |
| FCLA | M. flexor cruris lateralis pars accessoria | 0.0211 | 0.125 | 0 | 47.8 |
| ISF | M. ischiofemoralis | 0.0348 | 0.033 | 15 | 290 |
| PIFML | Mm. puboischiofemorales medialis + lateralis | 0.1273 | 0.089 | 15 | 389 |
| OM | M. obturatorius medialis | 0.457 | 0.055 | 25 | 2,160 |
| CFP | M. caudofemoralis pars pelvica (et caudalis) | 0.3069 | 0.108 | 15 | 778 |
| GL | M. gastrocnemius pars lateralis | 0.5706 | 0.12 | 20 | 1,269 |
| GIM | M. gastrocnemius pars intermedia | 0.2526 | 0.125 | 15 | 552 |
| GM | M. gastrocnemius pars medialis | 0.762 | 0.094 | 20 | 2,160 |
| FL | M. fibularis longus | 0.4791 | 0.081 | 20 | 1,570 |
| FDL | M. flexor digitorum longus | 0.1424 | 0.048 | 20 | 782 |
| FPPD3 | M. flexor perforans et perforatus digitorum 3 | 0.0822 | 0.025 | 30 | 798 |
| FPD3 | M. flexor perforatus digitorum 3 | 0.1605 | 0.017 | 35 | 2,220 |
| FPD4 | M. flexor perforatus digitorum 4 | 0.0955 | 0.026 | 20 | 992 |
| FHL | M. flexor hallucis longus | 0.0505 | 0.04 | 25 | 324 |
| EDL | M. extensor digitorum longus | 0.115 | 0.049 | 30 | 576 |
| TCf | M. tibialis cranialis (femoral head) | 0.165 | 0.045 | 25 | 474 |
| TCt | M. tibialis cranialis (tibial head) | | 0.045 | 25 | 474 |

**Table 3  Muscle wrapping surfaces assumed in the ostrich musculoskeletal model, with dimensions.** Examples are in Fig. 5.

| Muscle(s) | Location | Shape | r(x) | r(y) | r(z) | t(x) | t(y) | t(z) | Radius | Length |
|---|---|---|---|---|---|---|---|---|---|---|
| ILFB | Pelvis | Cylinder | 17.11 | 57.87 | −34.76 | 0.0309 | −0.0609 | 0.0622 | 0.095 | 1.000 |
| PIFML | Pelvis | Cylinder | 0.92 | −5.72 | −29.06 | 0.0020 | 0.0820 | 0.1000 | 0.170 | 0.500 |
| CFP | Pelvis | Cylinder | −8.51 | 0.41 | 35.68 | −0.0211 | 0.0722 | 0.1396 | 0.090 | 0.500 |
| ITC, ITM | Femur | Cylinder | −16.00 | 19.60 | 0.00 | 0.0086 | −0.0017 | −0.0067 | 0.020 | 0.500 |
| ITCR | Femur | Cylinder | −16.00 | −2.17 | −0.19 | 0.0395 | −0.0379 | −0.0034 | 0.020 | 0.500 |
| ITCR | Femur | Cylinder | 74.00 | 0.00 | −19.60 | 0.0086 | −0.0882 | 0.0181 | 0.055 | 0.500 |
| GL, GIM, FHL, FPD3 | Femur | Cylinder | 37.16 | 40.49 | −20.39 | −0.0197 | −0.2006 | 0.0799 | 0.015 | 0.200 |
| ILFBp | Femur | Torus | 51.43 | −21.08 | −22.24 | −0.0202 | −0.1827 | 0.0609 | 0.01[*] | 0.08[*] |
| FMTIM, FMTL | Tibiotarsus | Cylinder | 0.00 | 0.00 | 44.69 | −0.0014 | 0.0103 | 0.0093 | 0.038 | 0.500 |
| GL, GIM, GM | Tibiotarsus | Cylinder | 0.00 | 0.00 | 0.00 | −0.0058 | −0.4435 | −0.0090 | 0.040 | 0.150 |
| FP&PD3, FPD3, FPD4 | Tibiotarsus | Cylinder | 6.47 | −7.64 | 40.43 | 0.0031 | −0.4537 | 0.0090 | 0.030 | 0.200 |
| FDL, FHL | Tibiotarsus | Cylinder | 0.00 | 0.00 | 40.00 | −0.0014 | −0.4501 | 0.0001 | 0.030 | 0.500 |
| AMB2 | Tibiotarsus | Cylinder | 0.00 | 0.00 | 0.00 | 0.0250 | 0.0126 | −0.0012 | 0.038 | 0.100 |
| IC | Tibiotarsus | Cylinder | 0.00 | 5.00 | 0.00 | −0.0143 | 0.0098 | −0.0054 | 0.030 | 0.500 |
| EDL, TCf, TCt | Tibiotarsus | Cylinder | 3.37 | −15.47 | 0.32 | 0.0111 | −0.4530 | 0.0009 | 0.020 | 0.100 |
| FP&PD3, FPD3, FL | Tarsometatarsus | Cylinder | 0.64 | −6.17 | −1.51 | −0.0020 | −0.4296 | 0.0002 | 0.023 | 0.100 |
| FDL, FHL | Tarsometatarsus | Cylinder | −3.48 | −13.61 | −0.29 | 0.0025 | −0.4319 | −0.0024 | 0.022 | 0.100 |
| FPD4 | Tarsometatarsus | Cylinder | 0.00 | −35.00 | 0.00 | 0.0002 | −0.4326 | 0.0025 | 0.017 | 0.100 |
| EDL | Tarsometatarsus | Cylinder | −1.19 | −2.69 | 0.68 | 0.0029 | −0.4312 | −0.0152 | 0.023 | 0.100 |

| Muscle | Location | Shape | r (x) | r (y) | r (z) | t (x) | t (y) | t (z) | Radius (x) | Radius (y) | Radius (z) |
|---|---|---|---|---|---|---|---|---|---|---|---|
| IL | Pelvis | Ellipsoid | −2.79 | −1.44 | −100.93 | −0.0020 | −0.0020 | 0.0649 | 0.100 | 0.200 | 0.050 |
| IFE | Pelvis | Ellipsoid | 0.00 | 0.00 | 0.00 | −0.0010 | 0.0110 | 0.0302 | 0.090 | 0.060 | 0.060 |
| FCLP | Pelvis | Ellipsoid | −17.65 | 8.20 | −84.13 | −0.2258 | −0.0290 | 0.0532 | 0.075 | 0.500 | 0.050 |

**Notes.**

[*] the torus's outer and inner diameters, not radius and length, respectively

be equivalent to optimal fiber length ($l_o^m$; *Zajac, 1989*). We could thus calculate $A_{phys}$ as (e.g., *Alexander et al., 1979*; *Lieber & Boakes, 1988a*; *Brown et al., 2003b*; *Hutchinson, 2004*):

$$A_{phys} = m_{musc} \cos\theta (Ld)^{-1}. \tag{1}$$

Equation (2) then estimates $F_{max}$, maximum isometric force capacity:

$$F_{max} = 3.0 \times 10^5 \text{ m}^{-2} A_{phys}. \tag{2}$$

In which the constant is isometric stress under maximal activation (*Medler, 2002*; *Nelson, Gabaldón & Roberts, 2004*). Note that the musculoskeletal model varies $\theta$ with $L$ to maintain constant muscle thickness (*Zajac, 1989*). Muscle maximal contraction velocity was not calculated in the present model, but could be added (see *Smith et al., 2006* for example) for more complex simulations, using published fibre type data for ostriches (*Velotto & Crasto, 2004*) and ostrich or other avian muscle force–velocity data (e.g., *Nelson, Gabaldón & Roberts, 2004*). We also did not include measurements of tendon force–length data here, but either dimensionless estimates (*Zajac, 1989*) or specific measurements can

be added to future simulations. For simplicity, we chose to focus in this initial study on near-isometric muscle action and progress to more complex, dynamic parameters in later work.

Muscles were identified as in Table 2 following *Gangletal (2004)* and *Zinoviev (2006)* (see Appendix for additional details). For simplicity, we combined some muscle heads into single functional muscles in the model (cf. *Gangletal, 2004*: Table 1) and omitted some tendinous/minute muscles which were grouped with other similar muscles nearby or omitted in the case of M. popliteus (rotates fibula around tibia; *Fuss, 1996*); these simplifications are outlined in Text S1. However, some muscles (e.g., M. iliofibularis, M. iliotrochantericus caudalis) were large enough that separation into two heads was deemed important, as some heads might have very different flexor/extensor moment arms than more cranial/caudally-positioned ones.

## Limb muscle biomechanics: calculations and hypothesis testing

The musculoskeletal model was then imported into OpenSim (opensim.stanford.edu) software in order to take advantage of the programme's established analysis capabilities. OpenSim uses the 'virtual work' method (change of muscle–tendon unit length per unit joint rotation) explained by *Delp & Loan (1995)*, *Delp & Loan (2000)* and *Pandy (1999)* to compute muscular moment arms over a range of motion. Maximal muscular moments then can be estimated using muscle $F_{max}$ and potentially $l_o^m$ (see above and *Zajac, 1989*).

To test whether ostrich muscle moment-generating capacity is optimized to match peak loads during walking and running (our Question 1), we compared the results from estimated maximal muscle moments to experimentally-calculated internal and external moments (*Rubenson et al., 2011*), addressed in the Discussion. First, each muscle's maximal isometric muscle force ($F_{max}$) was multiplied by the flexor/extensor moment arm calculated by OpenSim (i.e., from the individual trials' limb joint angle input data and the model's resulting moment arm output data), for each pose adopted throughout the representative walking and running gait cycle trials (every 1% of gait cycle) to obtain the relationship between locomotor kinematics and isometric muscle moments. Second, OpenSim was used to calculate individual muscle moments directly, taking into account muscle force–length relationships (set as dimensionless in a Hill model as per *Zajac, 1989*), in order to provide a more realistic estimate of the variation of maximal moment-generating capacity throughout the same gait cycles. Both approaches were static, ignoring time/history-dependent influences on muscles. The second approach allowed non-isometric muscle action to be represented, but did not incorporate force–velocity effects, which would require a more dynamic simulation to resolve. Total extensor and flexor maximal moments were calculated in OpenSim as well as the net (extensor + flexor) maximal moment.

To determine if ostrich limb muscle moment arms peak at extended limb orientations or at mid-stance of locomotion (our Question 2), we used the model to calculate the mean moment arm of all extensor or flexor muscles across the full range of motion of each joint (estimated from osteological joint congruency as in *Bates & Schachner (2012)*) in

flexion/extension (set at constant values for mid-stance of running in other degrees of freedom), summed these mean moment arms, and divided that sum by the summed maximal moment arms for each muscle across the same range of motion (as in *Hutchinson et al., 2005*). We then inspected whether our representative mid-stance poses in walking or running matched maximal or minimal averaged moment arms corresponding to those poses.

To compare the degree of matching between muscle moment arms in our model and the experimental data of *Smith et al. (2007)* and *Bates & Schachner (2012)* (our Question 3), we obtained the published experimental and modelling data (KT Bates, provided by request), transformed their joint angle definitions to be consistent with our model definitions, and plotted the muscle moment arms vs. each joint angle with our moment arm data (also see Figs. S1–S4), restricting the other studies' ranges of motion to those presented in the original studies. For the knee and joints distal to it, in this study we focus only on flexor/extensor moment arms for simplicity and because the importance of long-axis and ab/adduction muscle (vs. passive tissue) moments at these distal joints is unclear, although our model could be adjusted to calculate those non-sagittal moment arms and moments.

## RESULTS

Here we present our data for addressing our main questions, proceeding in order with maximal muscular moments, maximal/minimal moment arms, and then general moment arm patterns compared with other studies.' The model is downloadable from the repositories at https://simtk.org/home/opensim and Figshare (*Hutchinson et al., 2015*) and can be manipulated in open source software OpenSim. Movie S1 shows the model animated through the representative running stride (from *Rubenson et al. (2007)*). Figure S5 shows the kinematic data for the walking trial vs. a larger dataset (not shown in *Rubenson et al. (2007)*, but obtained from the same methods and experiment). In that representative trial, the pelvic pitch angle is ~10° more tilted (pitched upward) than the mean—reasons for this isolated deviation are unclear but also very unlikely to influence our findings here. Both individual walking and running trials' data are supplied in the repository (*Hutchinson et al., 2015*).

### Maximal muscular moments

Our 3D ostrich limb model predicted how the maximal capacity to generate muscle moments should vary with limb orientation during walking and running (Figs. 6 and 7). Maximal flexor moments increase if force–length properties are ignored (treating all muscles as isometric). This indicates that most muscles in the model are at disadvantageously short fibre lengths during locomotion, with walking having a generally greater capacity for flexor moment generation (especially about the hip) than running. These curves do not change much across the gait cycle. The pattern for extensor moments is more complex. Peak capacity tends to be in late swing phase (reasonably consistent across all joints). Force-length properties here provide an advantage, presumably because the muscles are lengthened. Data during the stance phase do not support the hypothesis, regardless of assumptions about muscle-force length states, that postures used around mid-stance of walking or running optimize the moment-generating capacity of pelvic limb muscles in

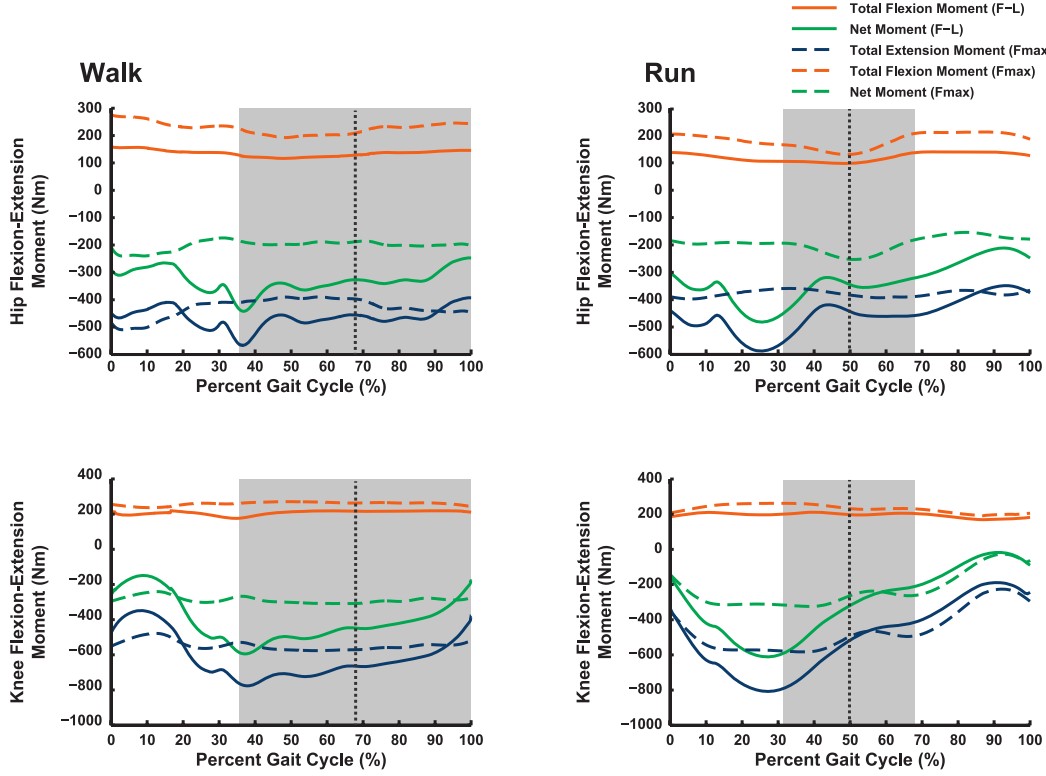

**Figure 6 Maximal muscle moments about proximal limb joints (hip and knee), for representative walking and running trials (see 'Methods').** "F–L" curves incorporate effects of muscle force–length properties into moment calculations; "$F_{max}$" curves only assume maximal isometric muscle stress and thus ignore F–L effects. The shaded area represents the stance phase, and the vertical dashed line is mid-stance (i.e., 50%).

ostriches: the maximal moments early or late in stance phase, and late in swing phase, are of similar or greater magnitudes. The relatively flattened shapes of most moment curves without force–length properties enforced ("Fmax"; dotted lines in Figs. 6 and 7) indicate that muscle moment arm variation across postures used *in vivo* during locomotion is a smaller contributor to moment generation than force–length properties ("F–L"; solid lines) in *Struthio*.

## Maximal/minimal muscle moment arms and limb orientation

Do ostriches' limb muscle moment arms peak at very extended limb orientations or at mid-stance of walking/running (Fig. 8)? We find that the mean hip extensor moment arms decrease from a peak at full extension as hip joint flexion increases, and the hip flexors behave similarly. However, knee and ankle moment arms each exhibit different patterns. The knee extensor and flexor moment arms tend to peak at moderate knee flexion angles (~60–90°), as do the ankle extensors (plantarflexors), but the ankle flexors have a near-plateau for most angles, quickly decreasing with extreme dorsiflexion (>100° ankle angle). When the poses that ostriches use during periods of peak limb loading (near mid-stance of walking and running; *Rubenson et al., 2007*) are compared against these patterns

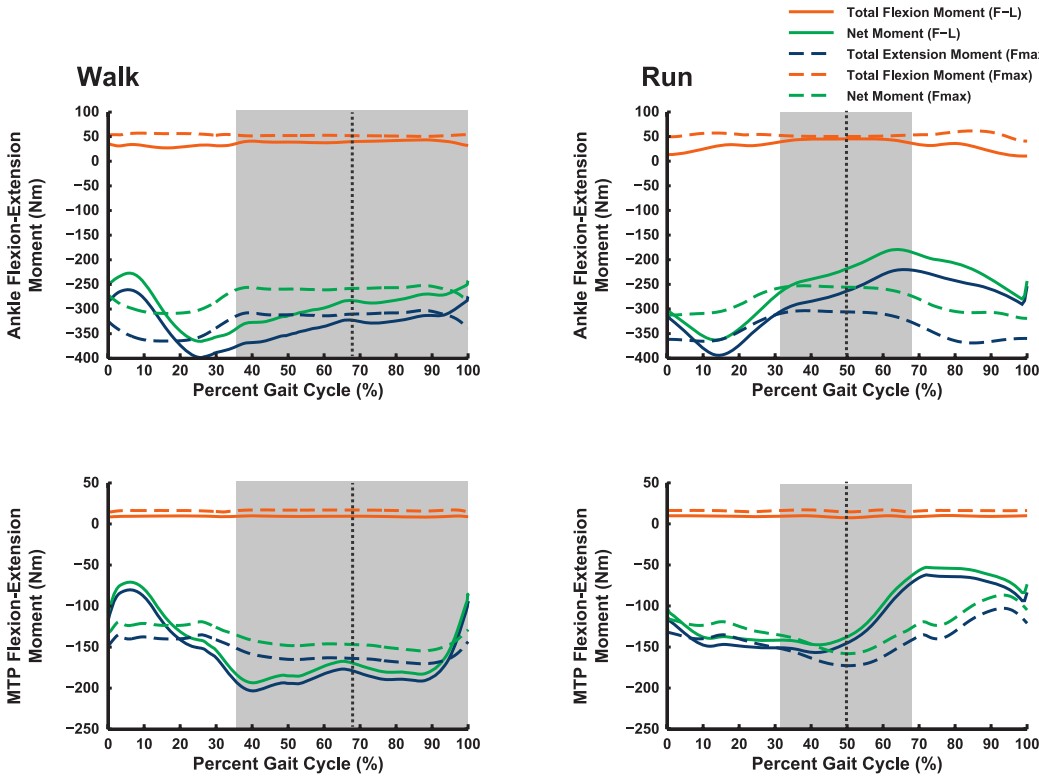

**Figure 7 Maximal muscle moments about distal limb joints (ankle and metatarsophalangeal MTP), for representative walking and running trials (see 'Methods').** See caption for Fig. 6.

(Fig. 8), it becomes evident that there is no clear optimization of muscle moment arms for supportive (large extensor or small flexor values) roles during these periods of potential biomechanical constraints. This is in agreement with the maximal moment data from Figs. 6 and 7. Hip extensors and flexors as well as ankle extensors are relatively far (~60–85% of maximal mean moment arms) from optimal values at mid-stance of walking and running. Knee extensor/flexor moment arms are closer to maximal values, especially for walking. However, the co-contraction of multiarticular hip extensor/knee flexors (e.g., ILFB, FCLP) against knee extensors would eliminate associated benefits—i.e., the ratio of peak knee extensor to peak knee flexor moment arms would have not have minimized the net knee extensor moments required at mid-stance of either walking or running. At moderate knee flexion values, both the capacity of muscles to extend and to flex the knee are near-maximal (Fig. 8).

## Moment arms: general trends and comparisons with prior studies

Figures 9–11 show our results for hip flexion/extension moment arms of ostrich muscles, with comparable data from *Smith et al. (2007)* and *Bates & Schachner (2012)* also plotted if available (abbreviated in this section as S.E.A. and B.A.S. respectively). Here we focus on the major findings. The two AMB muscles (Fig. 9) compare reasonably well among all three studies, showing a decrease of hip flexion moment arms at strongly flexed limb poses

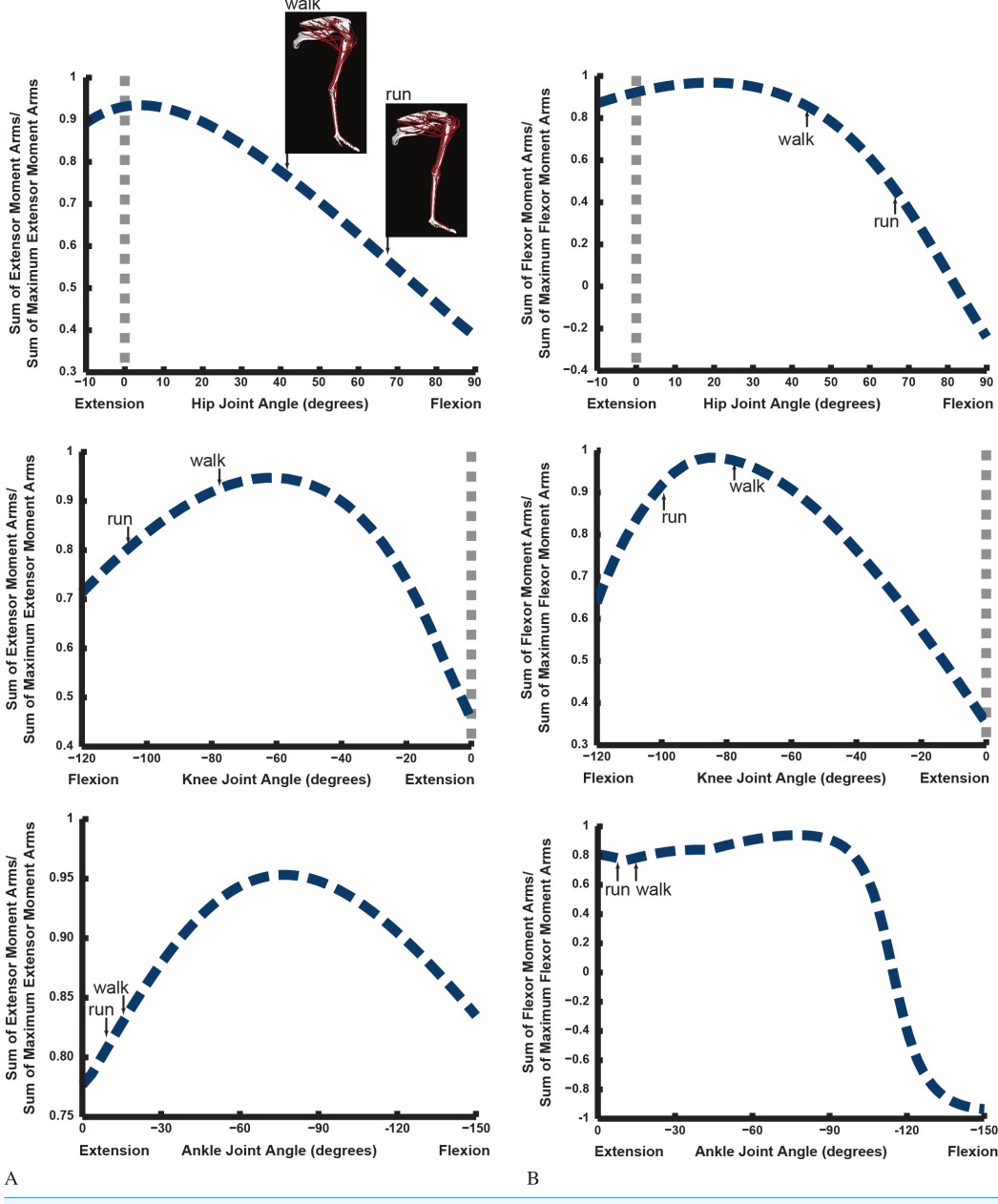

**Figure 8** Sum of extensor moment arms (A) or flexor moment arms (B) normalized by sum of maximal extensor or flexor moment arms, plotted against extension or flexion joint angle for the hip, knee and ankle joints (MTP joint data follow Fig. 20), with representative mid-stance limb poses for walking and running indicated.

and in some cases (our AMB1,2 and the AMB of B.A.S.) a switch from flexor to extensor action with flexion (∼30–90°). The IC muscles likewise have reasonably comparable results, but only our IC muscle switches action at extreme flexion. Our model agrees well with the data of S.E.A. and especially B.A.S. for the IL muscle, including its decreasing hip extensor moment arm with increasing hip flexion and a switch from hip extensor to flexor action at typical *in vivo* positions (∼40–70°). We have similar findings for the ILFB muscle,

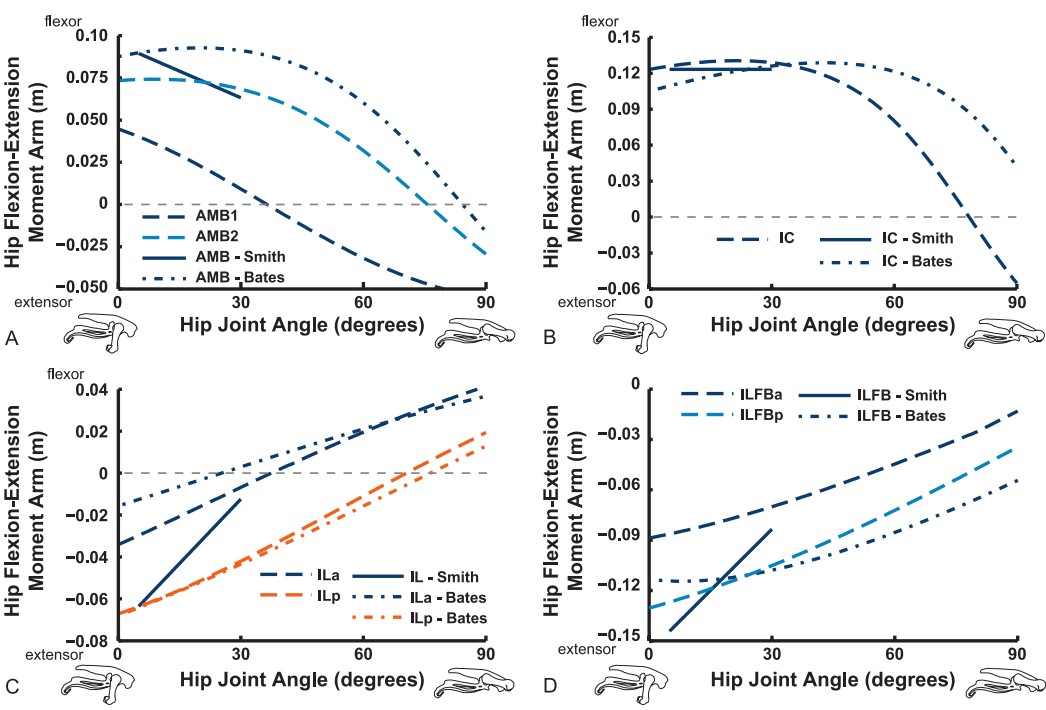

**Figure 9** Hip flexor/extensor moment arms plotted against joint angle for key proximal thigh muscles in our model, with corresponding data from *Smith et al. (2006)* labelled as "Smith" and from *Bates & Schachner (2012)* labelled as "Bates." Extreme extended/flexed right hip joint poses shown along the *x*-axis. Muscle abbreviations are in Table 2. Colours and line solidity are kept as consistent as feasible to reflect the study (e.g., Smith in blue solid lines) and muscle (e.g., reddish shades for parts of the AMB muscle in our data).

although no switch to hip flexor moment arms is observed in either of the two parts of this muscle in our model (S.E.A. and B.A.S. represented it as one part) (Fig. 9).

Uniarticular muscles acting about the hip joint consistently display flexor action for the IFE, IFI, ISF and OM muscles (Fig. 10). We find fair agreement among studies for the IFE (note confusion caused by misidentification of muscles in prior studies—see Appendix; the "IFE-Smith" in Fig. 10 is equivalent to our IFE and ITC), ITC, IFI, ITM and ITCR muscles' general changes of moment arms. Our IFE moment arm values are smaller than for S.E.A. and B.A.S. apparently because of the aforementioned identification issue (Fig. 10A shows our IFE plotted against S.E.A.'s IFE + ITC combined). Notably, the curves for the two parts of ITC in our data and those of B.A.S. are remarkably similar (and consistent with S.E.A.'s experimental data for their "IFE-Smith" as well as "ITC-Smith") despite the subjectivity inherent in partitioning this large muscle into two paths. These moment arms grade from flexor to extensor action with strong flexion (∼40–70°). A similar trend is evident for the ITM and ITCR muscles (but note the identification issues outlined in Appendix; S.E.A.'s "ITC" is actually the ITM, which their data otherwise lacks, so Fig. 10B compares their actual ITM ["ITC-Smith"] vs. our ITM). The antagonistic OM and ISF muscles concur less closely between the latter two studies, however, displaying more convex

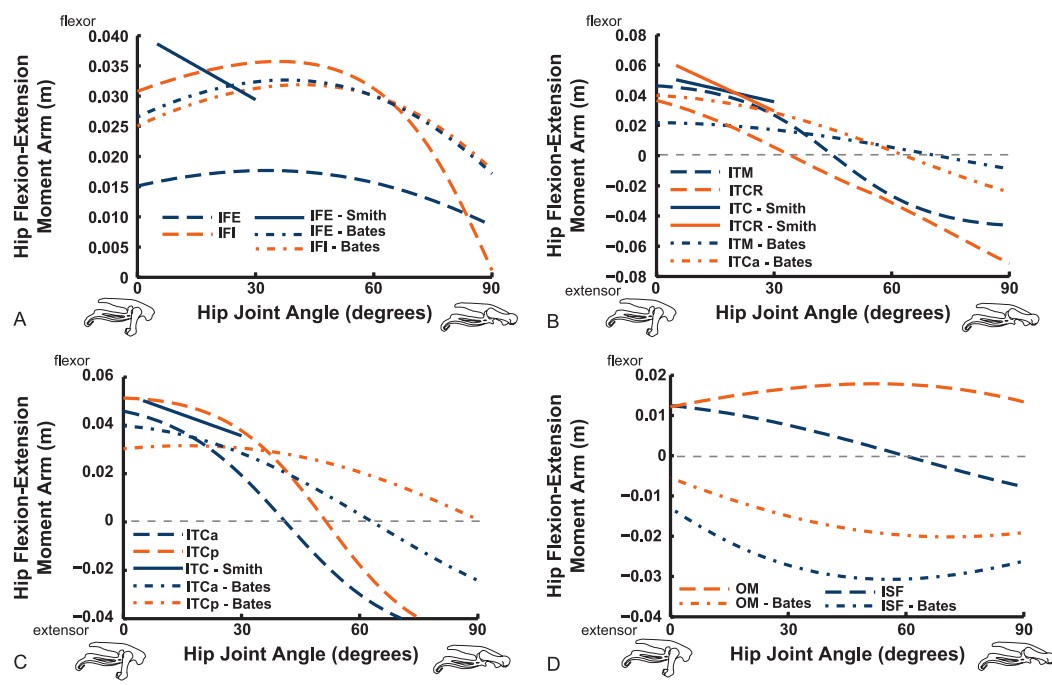

**Figure 10** Hip flexor/extensor moment arms plotted against joint angle for key proximal thigh muscles. See caption for Fig. 9. Dot-dashed lines represent "Bates" data here, whereas our data are in dashed lines.

curves tending to indicate hip flexor action in our data, with more concave, flattened arcs favouring hip extensor action in B.A.S. (Fig. 10).

The "hamstring," caudofemoral and adductor hip muscles uniformly display extensor action, befitting their more caudal paths relative to the hip, but agree less well among studies than the prior muscles (Fig. 11). Our data for the FCM, FCLP, CFP and PIFML muscles portray peak moment arms at low hip extension angles (~0–30°), decreasing with flexion away from these ranges. These trends qualitatively agree with the S.E.A. and B.A.S. data, but moment arm values tend to be substantially smaller in those data, especially for the FCLP and FCM muscles. Our PIFML data show less variation with joint angle than the S.E.A. and B.A.S. data because we had to constrain this muscle's path in 3D to avoid it cutting through bones or other obstacles in some poses. Note also how the S.E.A. results in general show strong changes with joint angles, whereas the more constrained muscle geometry of our model and B.A.S.'s results in more modest changes (Fig. 11).

Long-axis rotation (LAR; in Figs. 12 and 13) moment arms for hip muscles only allow comparisons between our data and those of B.A.S . Furthermore, considering that B.A.S. plotted these moment arms against hip flexion/extension joint angle (modified data shown; Karl T. Bates, pers. comm., 2015), we show them that way here but also plot them against hip LAR joint angle in the Supporting Information (Figs. S1 and S2); however, we do not discuss the latter results here. For the AMB1,2 muscles we find consistently weak, near-zero LAR action (lateral/external rotation), whereas B.A.S. showed a steeply decreasing hip medial/internal LAR moment arm as the hip is flexed
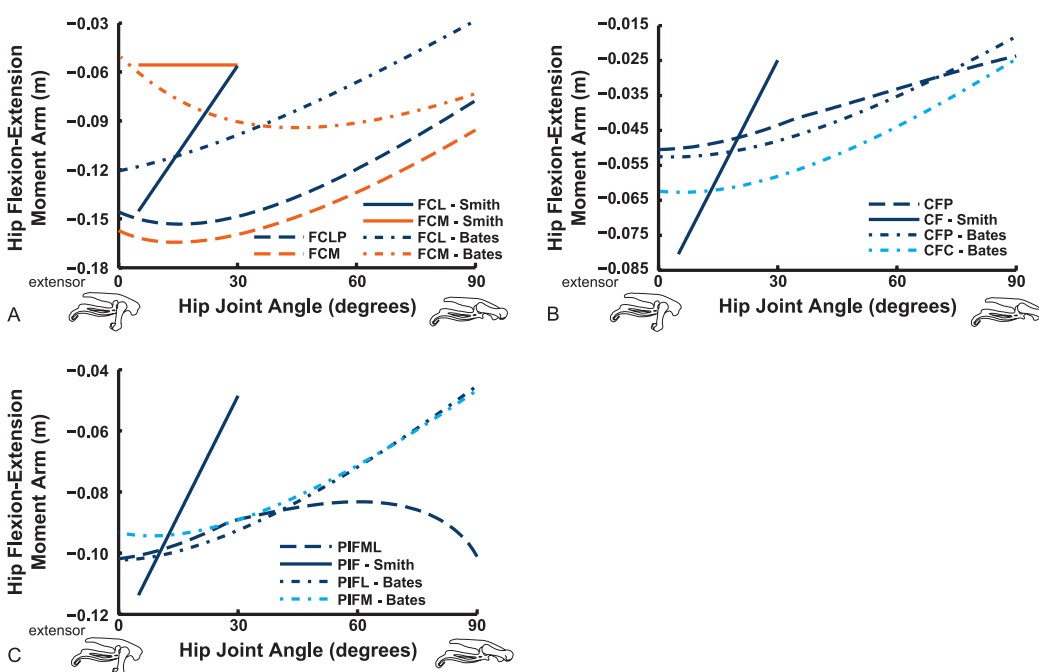

**Figure 11 Hip flexor/extensor moment arms plotted against joint angle for key proximal thigh muscles.** See captions for Figs. 9 and 10.

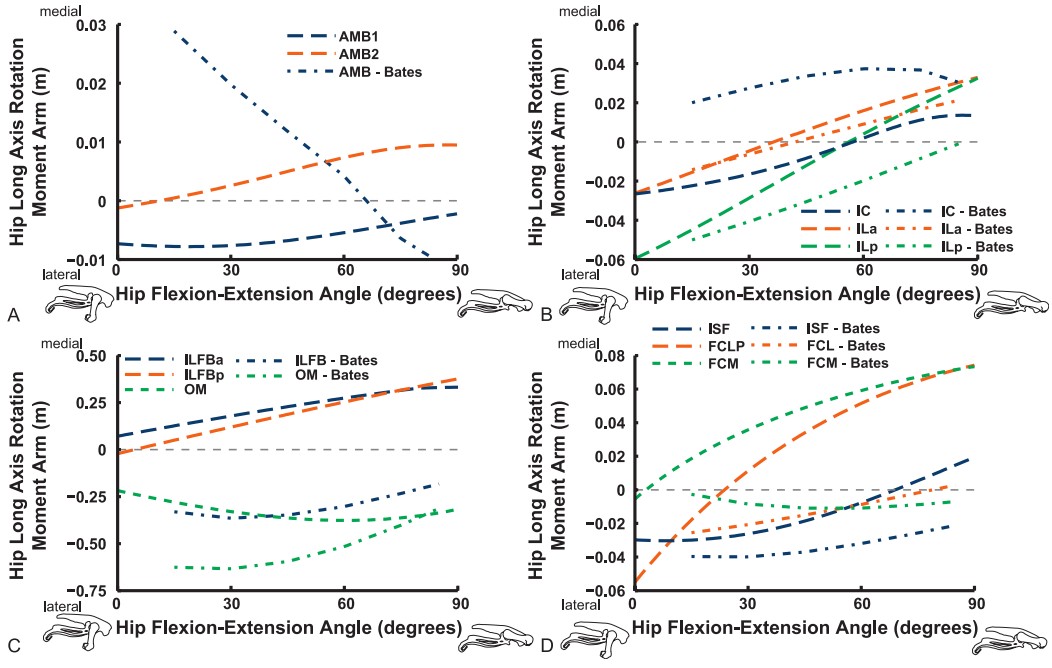

**Figure 12 Hip long-axis rotation (LAR) moment arms plotted against hip flexion/extension joint angle for key proximal thigh muscles.** See caption for Fig. 9.

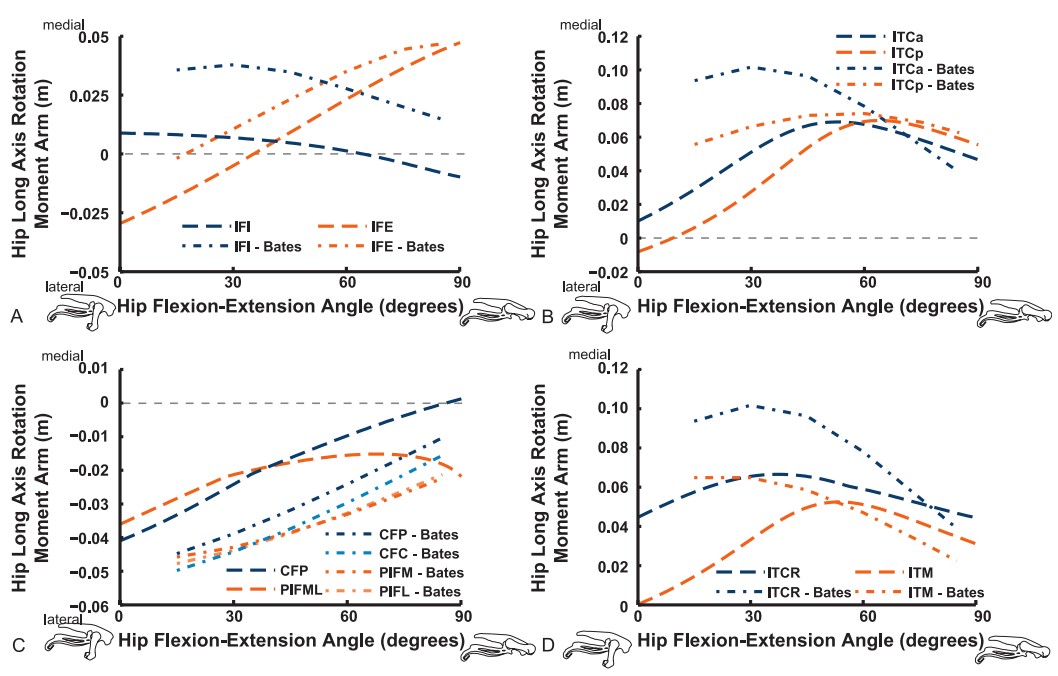

**Figure 13 Hip long-axis rotation (LAR) moment arms plotted against hip flexion/extension joint angle for key proximal thigh muscles.** See caption for Fig. 9.

(Fig. 12). In contrast, our IC and IL muscle data agree well with B.A.S.'s in having a shallow increase of the medial/internal LAR moment arm with hip flexion, although B.A.S.'s data much more strongly favour a medial rotator function for the IC muscle. Our results for the two parts of the ILFB muscle are very different from B.A.S.'s in trending toward stronger medial/internal rotation function as the hip is flexed, whereas B.A.S.'s favour lateral/external rotation. The results for the OM muscle have better matching between studies, indicating a lateral/external rotation action for this large muscle. Likewise, our ISF data and those of B.A.S. match fairly closely, with consistent lateral/external rotator action. The FCM and FCLP muscles have among the largest LAR moment arms for all muscles (∼0.08 m; also observed for our ILp muscle) in our data, but both muscles reduce their lateral rotator action with increasing hip flexion. In B.A.S.'s data a weaker, opposite (medial/internal rotator) trend with hip flexion was found for the FCM, whereas the FCL muscle maintained a small lateral/external rotator action (Fig. 12).

The uniarticular hip muscles' LAR moment arms of our model tend to switch less often (at *in vivo* hip joint angles ∼30–60°; e.g., Fig. S5) from medial to lateral rotation or vice versa (Fig. 13). The IFI, however, remains mainly as a weak medial rotator except at extreme hip flexion (>60°). B.A.S.'s data favoured stronger medial/internal rotation moment arms for the IFI but otherwise had a similar pattern. Our IFE muscle's data indicate a switch from lateral rotation into medial rotation near a 30° hip flexion angle, matched fairly closely by B.A.S.'s data. Our results for the two-part ITC muscle concur qualitatively with B.A.S.,' consistently having a strong medial/internal rotator action but smaller at more extended joint angles. As in B.A.S.'s data, but featuring smaller moment

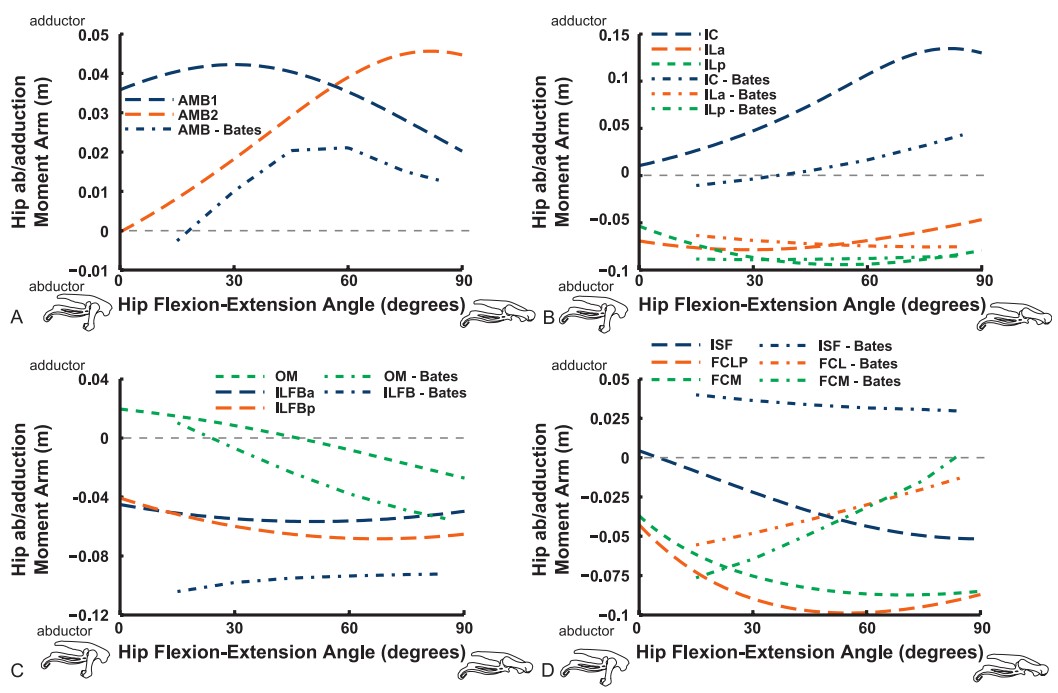

**Figure 14 Hip abduction/adduction moment arms plotted against hip flexion/extension joint angle for key proximal thigh muscles.** See caption for Fig. 9.

arms, our data show that the CFP and PIFML muscles have consistent lateral/external rotation action in ostriches; decreasing with increased hip flexion. The ITM and ITCR's medial rotator moment arms peak at hip angles of 30–60°, then decrease; a pattern qualitatively matched by B.A.S.'s data. (Fig. 13).

Abduction and abduction moment arms for the hip muscles show strong postural dependency like the LAR moment arms do (Figs. 14 and 15). Again, as for the LAR data above, we supply these data plotted against abduction/adduction hip joint angle in the Supporting Information (Figs. S3 and S4), , but we do not discuss those results here. The PIFML muscle has a discontinuity in its hip abductor moment arm (Fig. S4) in our model at extreme hip abduction angles (>−40°) but this is well outside normal *in vivo* abduction angles used (<25°; *Rubenson et al., 2007*). The two AMB muscles in our model have peak adductor moment arms at different flexion angles (∼30°and 80°), then decrease. B.A.S.'s data (modified data shown; Karl T. Bates, pers. comm., 2015) generally follow our AMB1 muscle's. Our IC muscle has a similar adductor moment arm curve as our AMB2, and a similar divergence from B.A.S.'s results, which remain close to a zero moment arm. Our IL muscle parts (ILa,p) agree well with B.A.S.'s, showing them to act as abductors. Both our ILFB muscle parts (ILFBa,p) have little variation in their hip abductor actions, whereas B.A.S.'s representation had a ∼100% larger moment arm but otherwise was similar. The OM muscle, which runs very close to the plane of the acetabulum, is an adductor at extended joint angles and an abductor at flexed angles in both our model and in B.A.S.'s data. Whilst the ISF muscle is almost exclusively a hip abductor in our model, it was exclusively an adductor in the B.A.S. model. The FCL and FCM muscles compare only

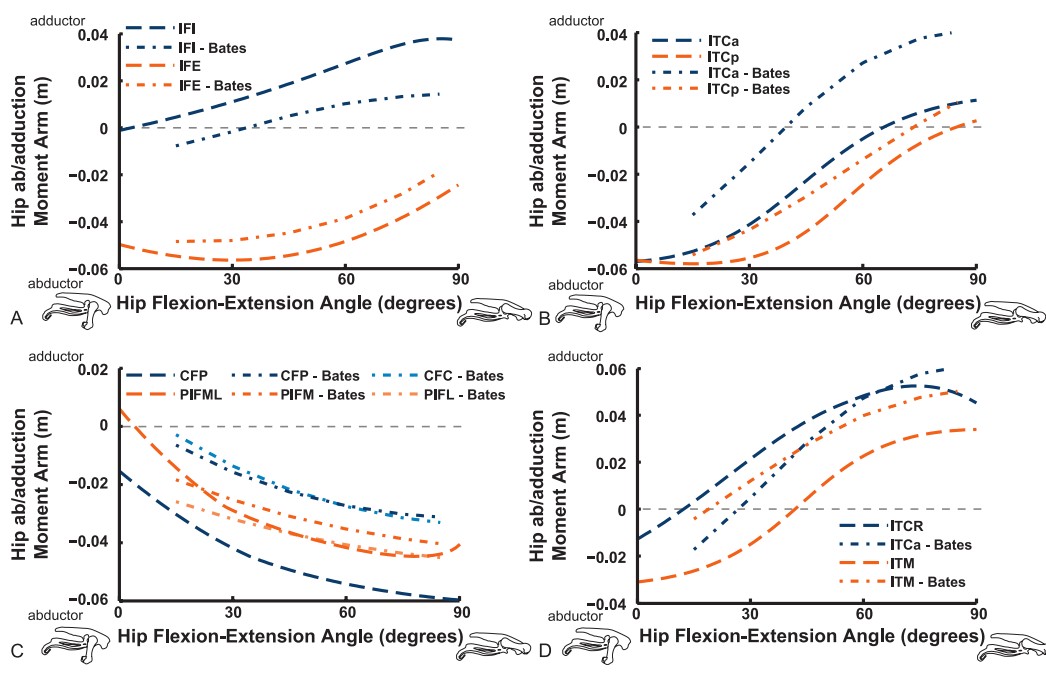

**Figure 15 Hip abduction/adduction moment arms plotted against hip flexion/extension joint angle for key proximal thigh muscles.** See caption for Fig. 9.

qualitatively between our data and B.A.S.'s, remaining as hip abductors. It is noteworthy that throughout the full ranges of hip motion we examined, most muscles would act as hip abductors; the dorsal AMB2 and IC muscles are the only consistently strong hip adductors (Fig. 14; Figs. S3 and S4).

Uniarticular "deep dorsal" and antagonistic muscles show similar trends as the above muscles for adduction/abduction capacities (Fig. 15). The IFI has weak adductor action, vs. a smaller, near-zero value (but similar trend) in B.A.S.'s data, whereas our data and B.A.S.'s agree well on the hip abductor moment arm of the IFE. Our representations of the ITCa/p muscle parts switch from abduction to adduction function as hip flexion surpasses 45–60°; B.A.S.'s model did this switch to a stronger degree. Postacetabular muscles such as the CFP and PIFML in our model are almost exclusively hip abductors, much as in B.AS.'s model. Finally, our results also generally have a good match to B.A.S.'s in the case of the ITM and ITCR muscles, which convert from abductor to adductor action at 10–40° hip angles (Fig. 15).

We only focused on flexion/extension moment arms for more distal joints, starting with the knee (Figs. 16 and 17). Good agreement between knee extensor moment arms for the AMB2 (dorsal) muscle is evident with the two other studies. The AMB1 (ventral) component only has data from our model (mainly a weak knee flexor), as does the IC (very weak knee flexor/extensor at flexed/extended angles). We estimate a larger knee extensor moment arm for the FMTL muscle but this is because of misidentification of part of that muscle in the S.E.A. and B.A.S. data (only a distal head was included in this muscle; see Appendix ). Similar differences of anatomical representation are likely explanations for the

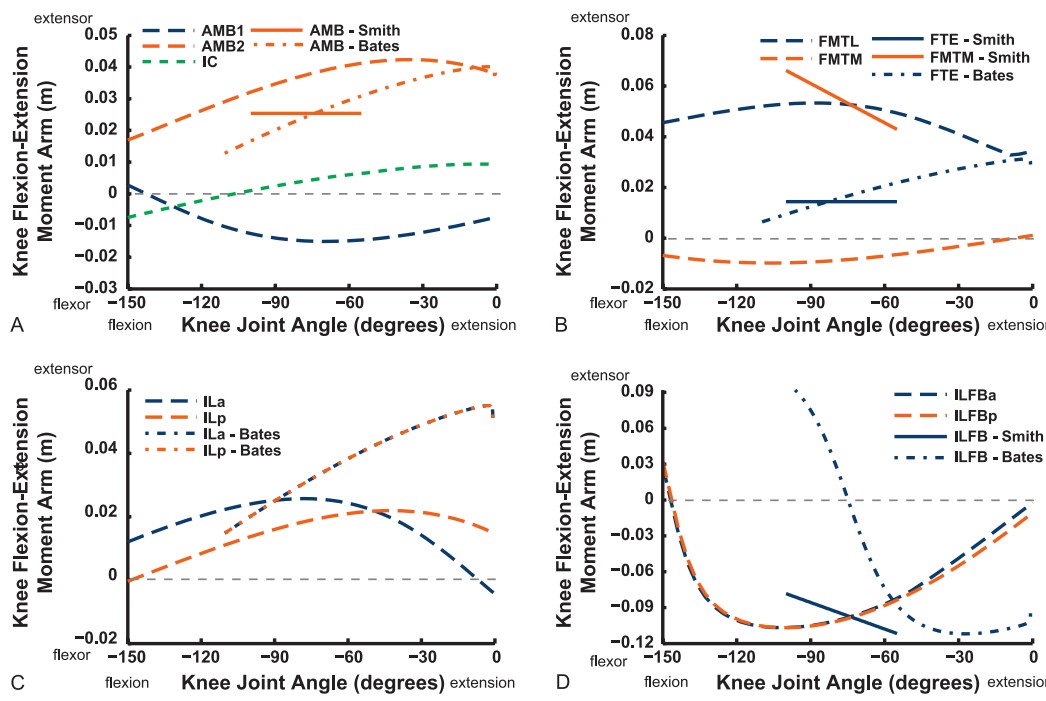

**Figure 16** **Knee flexor/extensor moment arms plotted against knee flexion/extension joint angle for key thigh muscles.** See caption for Fig. 9.

deviation between our result (weak knee flexor) and S.E.A.'s (strong knee extensor) for the FMTM (see 'Discussion'). Our model presents slightly different moment arms for its two IL muscle heads, peaking in extensor values at 30–90° flexion, whereas B.A.S. had identical moment arms increasing throughout extension.

We estimate the knee flexor moment arms as identical (peaking at 90–120° flexion) for the two parts of the ILFB muscle in our model, which match S.E.A.'s data well, whereas B.A.S. had moment arms switching from extensor to flexor at 70° of knee flexion, peaking at quite extended knee poses (Fig. 16) (see 'Discussion'). Other "hamstring" muscles (no comparable data for S.E.A. or B.A.S.) include the FCM and FCLP, which shift steeply from knee extensor to flexor moment arms at high flexion angles and then peak near 90° in its flexor moment arm value (Fig. 16). The FL muscle shows an almost mirror image pattern, acting as a knee extensor. Muscles running past the ankle joint (Fig. 17), such as the FPD3 and FPD4 groups, have a similar pattern to the FCM and FCLP at the knee, but the TCf muscle has almost no knee moment arm; consistently acting as a very weak extensor. Finally, parts of the gastrocnemius muscle group (e.g., GIM) reach peak knee flexor moment arms of about 0.07 m at intermediate knee flexion angles (60–90°). The data for S.E.A. and B.A.S. and for our GL muscle remain(ed) near smaller knee flexor values, with less postural variation (Fig. 17).

Ankle musculature displays fairly congruent patterns in our model and S.E.A. and B.A.S.'s data (Figs. 18 and 19). The TCf and TCt heads generally have an ankle extensor action, like the EDL muscle group does, albeit with some switches to extensor action

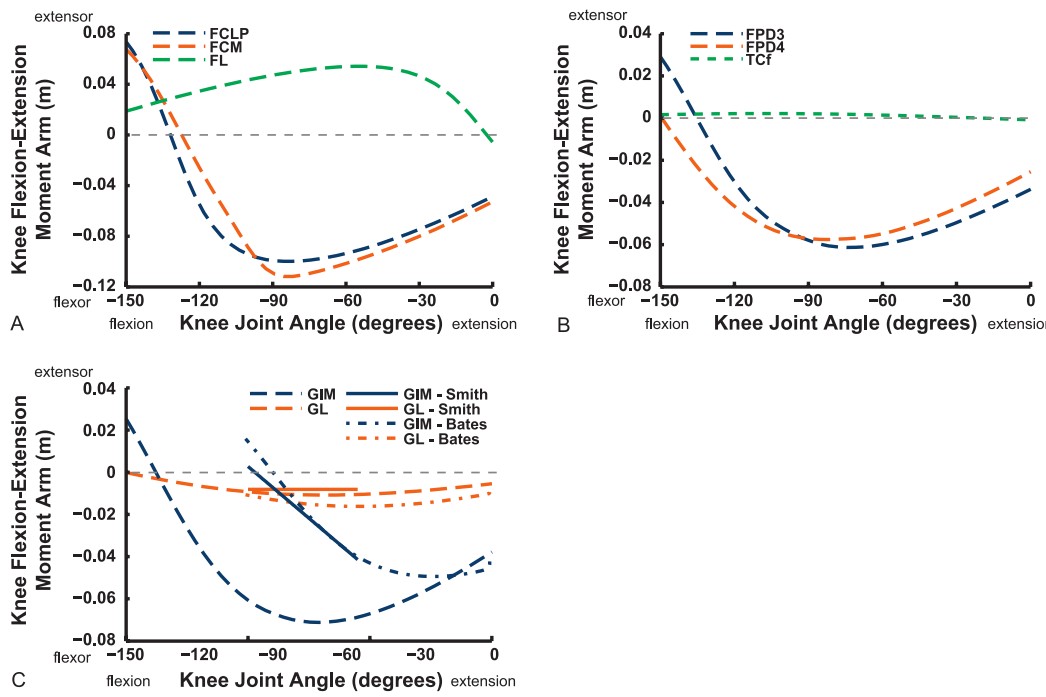

**Figure 17** Knee flexor/extensor moment arms plotted against knee flexion/extension joint angle for key thigh and distal knee muscles. See caption for Fig. 9.

with extreme (dorsi)flexion in the B.A.S. dataset (and our TCf). Surprisingly, ankle extensors reveal more variation: our FDL's ankle extensor moment arm is almost twice as large of that in the S.E.A. and B.A.S. data, showing little change with ankle posture, whereas the B.A.S. dataset exhibited a decreased moment arm with flexion. Our other digital flexor muscles (FPD3, FPD4) and those of S.E.A. display roughly similar values but opposite trends, increasing their moment arms with ankle flexion in our model. Our FL muscle's extensor moment arm is smaller than those of S.E.A. and B.A.S. The model of B.A.S. had a M. fibularis brevis (FB) muscle (Fig. 18), which is reduced to a ligament in *Struthio* and thus not included in our model; no studies have data for the ligamentous M. plantaris (*Zinoviev, 2006*). The extensor moment arms for our gastrocnemius muscles are all identical and fairly constant with ankle flexion, whereas the curves for the data of S.E.A. and B.A.S. increased steadily and tended to be larger (Fig. 19).

Digital flexor muscle moment arms all stay fairly constant (slight increase with extension of the MTP joint) in our model whereas they showed a stronger decrease in S.E.A.'s experiment (Fig. 20). Our EDL muscle has stronger moment arms than in S.E.A.'s data but a similarly shallow curve. Finally, our FL muscle exhibits digital flexor moment arms similar to those of the other digital flexors.

## DISCUSSION

The results of our combined experimental and theoretical approach show first that, while ostrich limb muscles are capable of generating large flexor and extensor moments about their limb joints during locomotion (Figs. 6 and 7), they do not seem to match maximal

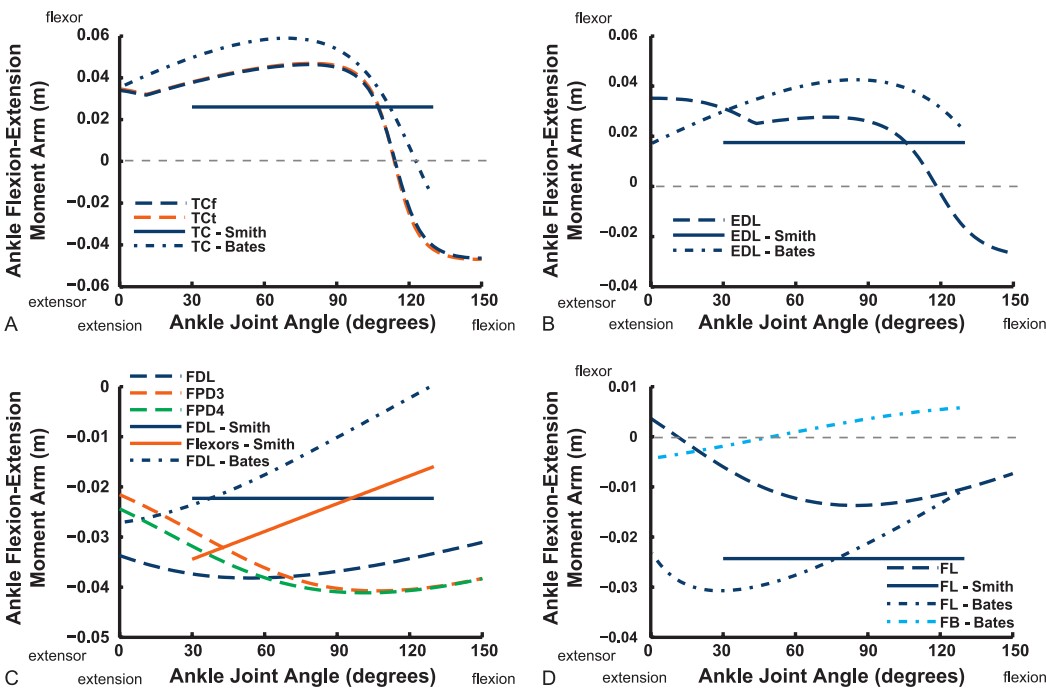

**Figure 18 Ankle flexor/extensor moment arms plotted against ankle flexion/extension joint angle for key muscles crossing the ankle.** See caption for Fig. 9.

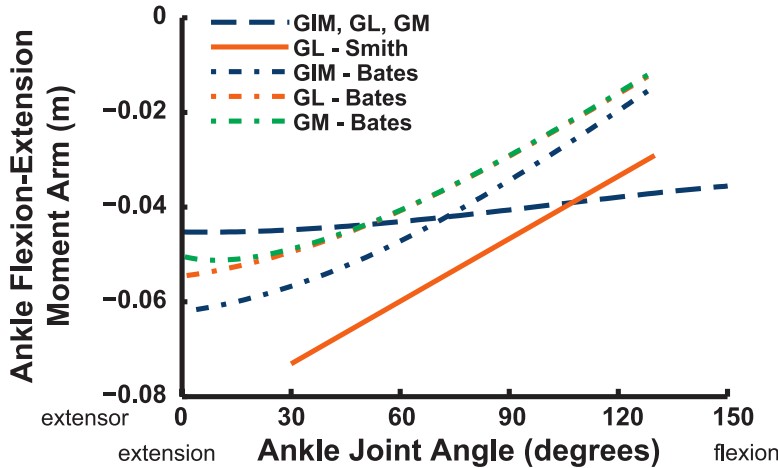

**Figure 19 Ankle flexor/extensor moment arms plotted against ankle flexion/extension joint angle for the M. gastrocnemius muscle group.** See caption for Fig. 9.

muscle moment-generating capacity with instants of peak loading in walking or slow running. Second, the moment arms of ostrich flexor/extensor muscles often change greatly with limb orientation, but they are not consistently matched to minimize the former and maximize the latter during key periods of weight support in locomotion (Fig. 8). Third, there is mostly reasonable consistency in three different studies of ostrich muscle moment arms (Figs 9–20), indicating at least fair repeatability with distinct methods, but still some

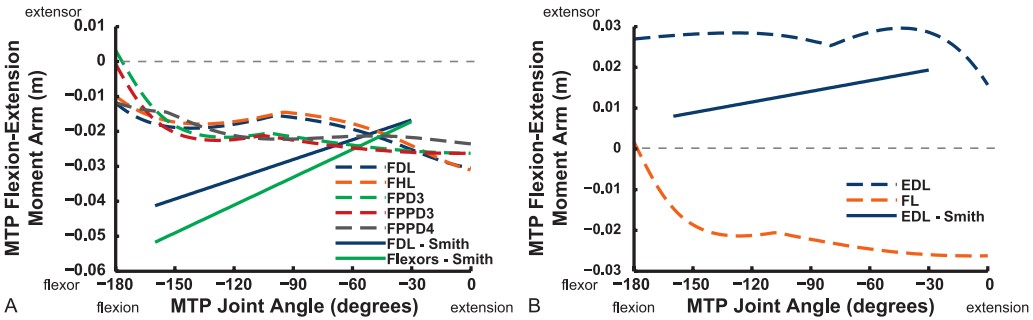

**Figure 20** Metatarsophalangeal (MTP) joint flexor/extensor moment arms plotted against MTP flexion/extension joint angle for digital flexors (A) and extensors, plus tendinous connection of M. fibularis longus (B). See caption for Fig. 9.

striking disagreements, especially in the little-explored area of non-flexor/extensor muscle mechanics. We explore these topics in more detail below and then consider broader issues related to our findings.

## Maximal muscle moments and kinematics

Our Question 1 asked, "Do ostriches adopt limb orientations during walking or running that optimize their capacity to generate maximal moments about the pelvic limb joints?" We find no convincing evidence of such optimization—maximal capacities to produce joint moments often peak either early in stance phase or during swing phase (Figs. 6 and 7). In both cases, net joint moments obtained from inverse dynamics analysis are low (*Rubenson et al., 2011*). Peak flexor moments (requiring extensor/antigravity muscle activity) occur at or near mid-stance in running ostriches (*Rubenson et al., 2011*: Fig. 7), reaching magnitudes $> -2$ N m kg$^{-1}$ (i.e., $> -150$ N m for our subject). Note that these magnitudes are far below the capacities of hip, knee and ankle muscles ($> 400$ N m; Figs. 6 and 7) but approach those of the digital flexors ($\sim 150$–200 N m; Fig. 7). On this basis, we infer that either passive tissues (including muscle passive force–length properties) play an important role in balancing moments about the tarsometatarsophalangeal joint in running ostriches (cf. *Haughton, 1864*; *Schaller et al., 2009*; *Schaller et al., 2011*), especially at faster speeds, or that muscle moment-generating capacity is near its limits for this joint in particular, even at slower speeds. Nevertheless, more proximal limb muscles seem further from their moment-generating limits.

In his classic biomechanical analysis of ostrich anatomy, *Haughton (1864)* assumed that "the greatest possible amount of muscular force shall be expended in straightening or unbending the legs," and thus that early and late stance respectively placed the greatest demands on these forces. Available data no longer support this notion, but there is no question that ostriches have muscle masses able to produce greater moments (and work) in extension than in flexion, as Haughton explained, but by a factor of about three times for the hip and knee rather than ten (*vide Smith et al., 2006*; *Smith et al., 2007*).

There are several potential explanations for our observations that lead us to a negative answer to our study's first question. First, we have only examined walking and slow

running. Near maximal speed, moment capacity and requirements around mid-stance might be more closely matched (e.g., *Hutchinson, 2004*), as forces surely increase. At a duty factor of 0.42, *Rubenson et al. (2011)* obtained peak vertical ground reaction forces of 1500–2000 N or about 2.17–2.89 times body weight (BW), whereas *Alexander et al. (1979)* estimated 2.7 BW peak forces for an ostrich at near top speed (duty factor 0.29). The latter study used an equation that probably underestimates peak forces for ostriches, as *Rubenson et al.*'s (*2011*) data show (peak forces are 16–55% greater than predicted from duty factor). Second, our present model is still static, not considering force–velocity or other dynamic interactions that would alter moment-generating capacities. It is possible that these parameters, or highly complex interactions (e.g., muscle moment arms and "power amplification"), could be more influential than the isometric and force–length properties that our model considers. Third, entirely different factors could determine locomotor and postural optimization, such as energetic costs or stability/manoeuvrability (e.g., *Daley & Usherwood, 2010*).

Comparison of our results with other studies of the relationship between limb orientation and muscle mechanics reveal a fourth potential explanation, that the optimization of anatomy, posture, physiology and other factors in locomotor dynamics could be highly species-, task-, limb-, joint- or muscle-specific. Lieber and colleagues (*Lieber & Boakes, 1988a*; *Lieber & Boakes, 1988b*; *Mai & Lieber, 1990*; *Lieber & Brown, 1992*; *Lieber & Shoemaker, 1992*) conducted an elegant series of studies that constitute a model system for addressing this issue. They elucidated that maximal moment production by the semitendinosus muscle in frog hindlimbs showed a strong dependence on muscle isometric force capacity and moment arms. Some of these studies found less dependence of moment production on joint angle-dependent moment arm values (e.g., *Lieber & Boakes, 1988a*; *Lieber & Boakes, 1988b*), but this dependency varied for the hip and knee joints (*Mai & Lieber, 1990*; *Lieber & Shoemaker, 1992*)—and might be expected to vary for other muscles, too. Indeed, the moment arm did not vary much with knee joint angle for the semitendinosus (e.g., 0.37–0.44 cm about knee, across 10–160° range of flexion/extension; *Lieber & Boakes, 1988a*: Fig. 6A) so this muscle could not contribute much variation to muscle moment production. One might predict more dependency of maximal muscle moment production on moment arms for muscles that have more variable moment arms, but this has not been conclusively determined. *Lieber & Brown (1992)* found that there was no simple relationship between muscle fibre length and moment arm in seven frog hindlimb muscles, with differences evident between muscles acting about the hip and knee, suggesting diverse adaptations to moment production demands.

The aforementioned studies' reviews of numerous others in humans, cats and other species likewise note some variability and uncertainty in what factors determine maximal moment capacity in limbs, so consensus has been elusive. Furthermore, *Brown et al.*'s (*2003b*) modelling/experimental study of horse forelimbs found that while the moment-producing capacities of flexor muscles were determined mostly by muscle properties (e.g., $F_{max}$ or muscle force–length), muscle moment arms could have greater effects on moments than those properties for some extensor muscles. *Young, Scott & Loeb (1993)* also

found variability from 50–100% of resting fascicle length in the amount of length change that cat ankle muscles used throughout their range of motion. Thus strict "sarcomere equivalency" (constant usage of maximal muscle range of motion; e.g., discussions in *McClearn, 1985*; *Lieber, 1997*; *Bates & Schachner, 2012*) is not expected, but an approximate "tuning" of moment arms to muscle fibre lengths (and joint ranges of motion) is expected at least in some cases (see *Sellers et al. (2013)* for an example of how such information could be useful). *Lieber & Shoemaker (1992)* explained how a greater muscle fibre length to moment arm ratio would cause muscle force–length properties to become less influential on muscle moment production. Therefore, in the long-fibred proximal muscles of ostriches and other birds that run closer to the hip and knee joints and thus have smaller moment arms (e.g., *Smith et al., 2006*; *Smith et al., 2007*), individual muscles' moment arms might be quite influential. This speculation has yet to be conclusively tested, let alone integrated into studies of whole limbs and locomotor dynamics.

Regardless, recent studies of the hindlimbs of mice (*Lieber, 1997*), rats (*Johnson et al., 2008*), and chimpanzees (*O'Neill et al., 2013*), as well as horse forelimbs (*Brown et al., 2003a*; *Brown et al., 2003b*) favour some optimization of locomotor tasks and muscle moment arms, as do broad comparative studies of elbow muscles by Fujiwara and colleagues (*Fujiwara, 2009*; *Fujiwara, Endo & Hutchinson, 2011*; *Fujiwara & Hutchinson, 2012*) and a comparative study of mammalian carnivores (*McClearn, 1985*). So far, however, general principles that extend across lineages, behaviours or anatomies remain elusive. We consider this ambiguity's effect on inferences about extinct taxa further below.

## Muscle moment arm-joint angle dependencies

Next, we posed Question 2: are the moment arms of limb muscles maximized (for antigravity/extensor muscles; or minimized in the case of antagonistic flexor muscles) at mid-stance of locomotion (optimizing weight support), or at highly extended limb orientations, as prior studies of dinosaurs inferred (e.g., *Hutchinson et al., 2005*)?

The peak extensor muscle moment arms that ostrich pelvic limb antigravity muscles have about the hip joint lie close to a completely columnar (i.e., vertical or 0°) hip angle, approximately 5°(Fig. 8), similar to prior results for *Tyrannosaurus rex* (*Hutchinson et al., 2005*). Ostriches, however, do not stand or normally move with such extended hip joints (*Rubenson et al., 2007*). We suspect this difference is because of their two orders of magnitude smaller body size (65+ kg vs. ~6,000+ kg) and hence the lack of necessity for extreme postural changes to maintain lower muscle stresses in order to maintain locomotor performance (*Biewener, 1989*; *Biewener, 1990*). However, ostriches may also have a greater importance for non-isometric muscle force–length properties in determining the limb orientation used (Figs. 6 and 7), as per the section above. Such speculations can be tested better once such physiological data exist for ostrich muscles. Our data also do not strongly support *Smith et al.*'s (*2007*) suggestion that hip extensor (or other muscle) moment arms are at peak values toward the end of stance phase (Figs. 6–8).

Overall, unfortunately the factors that determine limb orientation in locomoting ostriches, as the largest extant striding biped (and theropod dinosaur) available for study,

remain inconclusive, leaving the application of such principles to reconstructing limb orientations and locomotion in extinct theropods (e.g., *Hutchinson et al., 2005*; *Gatesy, Bäker & Hutchinson, 2009*) on shakier empirical and theoretical ground. However, this uncertainty is not cause for cynicism. It is an opportunity for future improvement, especially given the dearth of comparative studies that focus on how musculoskeletal mechanics relate to limb orientation, and the technical difficulties inherent to measuring or modelling muscle moment arms and other properties. Furthermore, quantitative biomechanical studies of extant or fossil organisms should still be considered a major step forward from past qualitative, intuitive or subjective functional studies.

### How accurate and repeatable are estimates of ostrich limb muscle moment arms?

Our study's Question 3 dealt with a methodological comparison among the three main studies of ostrich pelvic limb muscle moment arms. Agreement seems fair overall, especially for flexion/extension actions. However, several main messages emerge from our comparisons, some of which were also voiced by the other two studies of ostrich pelvic limb moment arms (*Smith et al., 2007*; *Bates & Schachner, 2012*; here "S.E.A". and "B.A.S".). Circumstantial support for all three methods' accuracy additionally comes from tendon travel measurements of cranial and caudal parts of the IL muscle in guineafowl by *Carr et al. (2011)*. General patterns (their Fig. 7) for the IL moment arms about the knee (concave arc, peaking ∼100° knee angle in flexion) and the hip (increasing with extension) agree reasonably well with these three ostrich studies (Figs. 12 and 16). However, all moment arms for the ostrich IL muscle infer a switch to hip flexor action in strongly flexed poses, and little or no levelling off of the moment arm curve at strong hip flexion angles.

Key areas of disagreement between our results and those of B.A.S. and/or S.E.A. include occasionally major differences in if, or how, muscles switch between flexion and extension (e.g., the AMB1 and AMB2, IC, ILFB about the hip; Fig. 9–11), whether certain muscles are flexors or extensors (e.g., the OM; see "Implications for ostrich limb muscle function" below), or the absolute magnitudes or relative trends in the data (e.g., our near-constant moment arms about the ankle for the FDL and gastrocnemius muscles; Figs. 18 and 19; and for the digital muscles, Fig. 20). We also found some differences in LAR and ab/adduction moment arms about the hip for B.A.S.'s data, but these are likely explained by differing muscle paths (e.g., via points and wrapping); see Figs. 12–15. *Bates & Schachner (2012)* acknowledge that their model could not use both via points and wrapping surfaces for the same muscle. This limitation explains the switch of their ILFB knee moment arm from flexor to extensor with knee flexion (unlike *Smith et al.*'s (*2007*) data); our model only exhibits this switch at extreme knee flexion (∼150° vs. 90°; Fig. 16). Similarly, our FDL's ankle extensor moment arm was nearly constant (Fig. 18), as in Smith's data, but the modelling limitation might explain why *Bates & Schachner*'s (*2012*) moment arm curve showed a stronger decrease with ankle flexion.

Contrastingly, the "M. femorotibialis medialis" (see Appendix ; equivalent to part of our FMTL; Fig. 16, "FTE-Bates") muscle's moment arm increased with knee extension in B.A.S.'s model, following a pattern similar to other knee extensors,' but S.E.A. found a

progressive decline of that muscle part's ("FMTM-Smith", Fig. 16) knee extensor moment arm with knee extension. By comparing homologous muscles, it is evident that S.E.A.'s experimental data ("FMTM-Smith", Fig. 16) match our model somewhat (FMTL; i.e., solid red and dashed blue lines in Fig. 16B). Furthermore, the two distal parts of FMTL ("FTE" in Fig. 16; B.A.S. and S.E.A. data) match each other somewhat. However, only our model represents the anatomy and function (weak knee flexion) of the actual medial head of M. femorotibialis (FMTM; Fig. 16), which was not distinct in the data of B.A.S or S.E.A. Thus, a combination of incorrect classification of muscles (Appendix ) and methodological differences explains divergence between some of the three studies' results. We noted similar problems with misidentifications (IFE, ITC, ITM muscles; Appendix ) above in describing the results shown in Fig. 10.

We raise the point here of the mistaken notion that tendon travel estimates of moment arms are free of errors in joint centre estimation, which is often repeated (e.g., *Smith et al., 2007*; *Channon et al., 2010*). This is only partly correct—by using the virtual work principle that moment arms are equal to the change in musculotendinous length ("tendon travel") per unit of joint rotation, tendon travel experiments do minimize some errors in measuring moment arms, but still require joint centre estimates to calculate joint rotations, forming the traditional *x*-axis of moment arm vs. joint angle plots. Admittedly, all studies may carry this burden of error; our model is different in using empirically measured 3D joint axes. Yet typical tendon travel experiments still introduce another error, by introducing inevitable kinematic cross-talk between flexion/extension movements and motions about other planes (i.e., limb joints that are manually flexed/extended through a range of motion will also involve some motion in LAR and ab/adduction—see *Rubenson et al., 2007*). Only the most rigorously constrained studies, which measure joint axes and constrain motions to strictly flexion-extension planes, avoid this problem. Models like ours and B.A.S.'s can explicitly avoid it. Our model's data show that, because moment arms of muscles co-vary with all joint angles (flex/extension, ab/adduction, long-axis rotation) (Figs 12–15, S1–S4), changes in multiple joint angles at once (i.e., kinematic cross-talk) will tend to produce different moment arms than changes in one angle at a time.

*Bates & Schachner* (*2012*: p.1342) inferred that, because their model's outputs matched experimental data, "predicted abduction/adduction and long-axis rotation moments are good estimates." We find some important differences between our results and those of either or both studies that somewhat weaken this inference (Figs. 12–15). However, we reiterate and celebrate that all three studies discussed here obtain broadly similar results for most muscles, in particular emphasizing that moment arms are not constant for most muscles. Furthermore, *Smith et al. (2007)* and *Bates & Schachner (2012)* noted limitations similar to those mentioned above.

Some comparative data exist from studies of other ratites or more restricted datasets for ostriches. *Hutchinson (2004)* provided estimates weighted by physiological cross-sectional area ($A_{phys}$) of "antigravity" (extensor) muscle moment arms for a 2D model of an ostrich, using the same specimen and hence congruent with our results. *Alexander et al. (1979)* created a simple 2D geometric model of an ostrich whose limb dimensions were similar

to ours (cf. their Table I and our Table 1), obtaining comparable muscle moment arm measurements: femorotibialis 5.1 cm about the knee (ours ∼5 cm), gastrocnemius 3.8 cm about the ankle (ours ∼4 cm), and digital flexors 3.2 cm about the MTP joint (ours ∼2 cm). This ostrich was 64% of the body mass of ours but had muscle masses about 85% of ours, helping to explain its ∼30% greater $A_{phys}$ and $F_{max}$ values (in addition, surely, to shorter muscle fascicles; Eqs. (1) and (2)), which are differences most likely attributable to its wild-caught status as opposed to our specimen's farm provenance.

*Goetz et al. (2008)* created a musculoskeletal model of an emu using a procedure grossly similar to ours, and obtained similar results—e.g., ILFB muscle having the largest hip extensor moment arm (but see Appendix ). *Troy, Brown & Conzemius (2009)* assumed that only the IFE, ITM, ITCR and PIFML muscles (homologous to ours) would resist hip adduction in their simplified emu model, but our analysis reveals that several more hip abductors exist, namely the IL, ILFB, FCM/L and CFP muscle groups (Table 4; Figs. 14 and 15). See *Lamas, Main & Hutchinson (2014)* for more consideration of emu pelvic limb muscle functions.

There are anatomical and methodological reasons to suggest that musculoskeletal modelling approaches (this study and *Bates & Schachner (2012)*) can sometimes be more reliable than tendon-travel-based empirical data (e.g., *Smith et al., 2007*; also possibly *Carr et al., 2011*). First, the typical experimental method, as applied to date with birds, removes surrounding muscles and replaces whole 3D muscle bellies with quasi-2D strings (often lacking key "via points" or wrapping surfaces) that must alter the lines of action and thus moment arms of the *in situ* muscles. The potential for such alteration is experimentally testable, but existing studies of moment arms in various species (especially humans) already give strong precedent for this inference: analogous comparisons of "straight-line" 2D measurements (e.g., *Jensen & Davy, 1975*) vs. 3D medical imaging or *in situ* measurements (e.g., *Young, Scott & Loeb, 1993*; *Arnold & Delp, 2001*) vs. anatomically-realistic 3D musculoskeletal models (*Delp et al., 1999*; *Arnold et al., 2000*; *Kargo & Rome, 2002*; *Brown et al., 2003b*; *Burkholder & Nichols, 2004*; *Blemker & Delp, 2006*; *O'Neill et al., 2013*) show better agreement between the latter two methods, and areas where simple 2D measurements are imprecise or misleading. To be fair, poorly designed musculoskeletal models can have the same problems—faithful representations of anatomical geometry are vital for both methods.

*Bates & Schachner* (*2012*, their electronic supplementary material) note that their model obtained an "extremely close match to the experimental data, particularly given the level of intra-specific variation present in muscle moment arms for ostriches." *Smith et al. (2007)* did attribute some of their experimental variation to individual differences in morphology that might alter moment arms, but as they noted some of the variation may have instead been due to experimental error. *Young, Scott & Loeb (1993)*, using a rigorous, validated *in situ* experimental apparatus for muscles crossing cat ankle joints, still found some individual variability that they attributed to muscles tethered by soft tissues as opposed to those restricted by bony wrapping surfaces or travelling in simple straight lines without any obstacles. We caution that these different sources of error need to be weighed

**Table 4 Muscle actions, following results from Figs 9–20, to describe the major 3D potential functions of each ostrich pelvic limb muscle.** Blank cells indicate the muscle does not cross or act about the joint. "+" signs added to classifications indicate a major potential role in these functions based upon moment arm and muscle relative size (i.e., moment generation capacity), subjectively assessed. "/" combinations (F/E; M/L; AB/AD) indicate a strong sensitivity of muscle moment arm, and hence action, to joint angle. Annotation with an asterisk indicates a potential role for intrinsic stabilization about that axis of motion (see Discussion). "()" indicates that our model's single origin for each muscle (or part thereof) did not allow such an action, but sub-parts of those muscles might have such actions if modelled in more detail.

| Muscle | Hip F/E | Hip LAR | Hip Ab/Ad | Knee F/E | Ankle F/E | MTP F/E |
|---|---|---|---|---|---|---|
| IC | F+* | M | AD+ | F/E | | |
| ILa | F/E | M/L | AB+ | E+ | | |
| ILp | E+ | M/L | AB+ | E+ | | |
| AMB1 | E* | L | AD | F | | |
| AMB2 | F* | M/L | AD | E | | |
| FMTL | | | | E+ | | |
| FMTIM | | | | E | | |
| FMTM | | | | F | | |
| ILFBa | E | M | AB | F+ | | |
| ILFBp | E+ | M | AB | F+ | | |
| ITCa | F/E* | M+ | AB/AD | | | |
| ITCp | F/E* | M+ | AB/AD | | | |
| IFE | F | M/L | AB | | | |
| ITM | F/E* | M | AB/AD | | | |
| ITCR | F/E* | M+ | AB/AD | | | |
| IFI | F | M/L | AD | | | |
| FCM | E | M | AB | F | | |
| FCLP | E+ | M+ | AB+ | F | | |
| FCLA | E | M | AB | | | |
| ISF | F/E* | L | AB | | | |
| PIFML | E | L | AB | | | |
| OM | F+ | L+ | AB/AD* | | | |
| CFP | E | L | AB | | | |
| GL | | | | F | E+ | |
| GIM | | | | F | E | |
| GM | | | | (F/E) | E+ | |
| FL | | | | E | F* | F |
| FDL | | | | | E+ | F+ |
| FPPD3 | | | | (F/E) | E+ | F+ |
| FPD3 | | | | (F/E) | E+ | F+ |
| FPD4 | | | | (F) | E+ | F+ |
| FHL | | | | (F) | E | F |
| EDL | | | | | F+* | E+ |
| TCf | | | | 0 | F+* | E+ |
| TCt | | | | | F+* | E+ |

**Notes.**

Classifications: E, extensor; F, flexor; M, medial (internal) rotator; L, lateral (external) rotator; AB, abductor; AD, adductor; 0, no moment arm per se despite crossing the joint.

separately; individual variability is an interesting biological reality, but experimental error needs minimizing. We suspect that some of the issues raised above about unrealistic representation of musculoskeletal anatomy in tendon travel experiments might be due to experimental error. *Bates & Schachner (2012)* also raised the latter issue, finding that trends evident in *Smith et al.*'s (*2007*) moment arm data would leave the hip extensors too weak to balance the hip joint moments during stance phase, with a premature shift to hip flexor action. Our results (Figs. 9–20) indicate that tendon travel measurements of moment arms often exhibit steeper changes with joint angle in some cases, or constant moment arms where models show good reason to suspect variable moment arms with joint angle.

*O'Neill et al.*'s (*2013*; also *Holowka & O'Neill, 2013*) musculoskeletal modelling and experimental analysis of chimpanzee pelvic limb moment arms came to similar conclusions as we have here, concerning model vs. tendon travel estimates of moment arms (see also *Brown et al., 2003a*; *Brown et al., 2003b* for horses). Importantly, they noted that non-linear trends in moment arm vs. joint angle data are to be expected, so tendon travel studies that enforce the fitting of largely linear (or simple polynomial) curves to their data may introduce errors. *O'Neill et al. (2013)* also cautioned that (p. 3721) "the 3D orientation of the pelvis and thigh segments during tendon-excursion experiments can have a significant effect on moment arm metrics." *Channon et al. (2010)* used cubic spline functions to characterize gibbon hindlimb muscle moment arms, presenting an improvement over the method used for ostriches by *Smith et al. (2007)* and considering some of the potential errors in tendon travel methods. Tightly controlled experiments with cats (*Young, Scott & Loeb, 1993*) have shown good agreement with model-based moment arm data (*Burkholder & Nichols, 2004*).

Nonetheless, there is room for improvement in all methodologies. In particular, human experiments have shown that moment arms may depend on muscle activation levels (*Maganaris, 2004*; *Tsaopoulos et al., 2007*), a phenomenon that few moment arm studies have assimilated. Numerous studies have also shown that sarcomere or fibre length changes in real 3D muscles, as opposed to line segments, may be highly heterogeneous and complex (e.g., *Blemker, Pinsky & Delp, 2005*; *Carr et al., 2011*).

The accuracy of avian musculoskeletal models remains unresolved. Studies of moment arms for ostriches and other avian species badly need stronger validation tests—ideally 3D, *in vivo* and across a range of behaviours—to test where different methods succeed or fail. *O'Neill et al. (2013)* noted that model-based estimates of moment arms should be less sensitive to the origins of muscles than to the insertions, and given that those insertions tend to be more easily circumscribed in ostriches (as in chimpanzees), a focus on improving the placement of muscle insertions could be powerful. At present, there is no published gold standard measurement that the aforementioned studies can be compared against, and thus it is unclear which estimates of moment arms are truly more accurate and reliable than others. Our judgements above might prove to be incorrect. We assume here, except where noted, that our moment arm estimates are generally an improvement over previous studies' because they are 3D, based on precise, subject-specific

anatomical measurements of a single cadaver *in situ*, and incorporate modern data on the 3D complexity of avian limb joint axes. However, our assumption of improved accuracy demands a test against a gold standard, with clear criteria for what a "good" agreement between moment arm curves is; a question that no studies (including ours) have answered.

## Model assumptions and potential refinements

Some simplifications of joint systems were necessary in our model but could be improved with later iterations. The tibio-fibular articulation is slightly mobile in ostriches (*Fuss, 1996*) and other birds, but we maintained it as an immobile joint. Likewise, the (proximal; see *Chadwick et al., 2014*; *Regnault, Pitsillides & Hutchinson, 2014*) patella surely translates (and perhaps rotates) during knee flexion/extension in birds as in humans (e.g., *Walker, Rovick & Robertson, 1988*; *Suzuki et al., 2012*) but we maintained it in the same resting position (with respect to the femur), represented simply by a wrapping surface. Adding such translation would influence the moment arm curves for knee extensor muscles. The intertarsal (ankle) joint's motions during swing phase (extreme dorsiflexion) seemed unrealistic, laterally rotating the tarsometatarsus to a seemingly disarticulated position (see Movie S1), but we kept this as-is in the model rather than invent a subjective solution, as it would have minimal influence on our results here and maintained strict fidelity to our anatomical and kinematic data. Future implementations of the model emphasizing ankle joint mechanics (especially in the swing phase when dorsiflexion is prominent) may need to adjust these kinematics. The proximal interphalangeal joint of digit III was kept immobile in the model because our kinematic data lacked its angular motions, but the model has the capacity to allow the joint to flex and extend if desired (Table 1; *Hutchinson et al., 2015*), and could involve internal mechanisms such as those described by *Schaller et al. (2011)* if necessary for research questions addressed with it.

Our model's muscles were simplified, as the Methods and Text S1 Text explain. Our digitizing procedure, performed in 2002, was simplistic (similar to that of *Burkholder & Nichols, 2004*), whereas more recent techniques have fused CT and MRI imaging modalities to produce quite accurate and complex 3D musculoskeletal models (e.g., *Zarucco et al., 2006*; *Harrison et al., 2014*). Real muscles have complex 3D structure but we have simplified them into basic Hill model muscles of 2D structure. Internal tendons were observed in some muscles (e.g., M. iliotrochantericus caudalis, Mm. gastrocnemii, many digital flexors; *Gangletal, 2004*). The Hill model does not discretely represent these features, which can affect muscle forces and gearing. Ligaments and other passive tissues were not represented in our model, and these would be particularly important features to consider in a complete dynamic model, as *Haughton (1864)* suggested and *Schaller et al. (2009)* demonstrated experimentally. Finally, a test of the validity of our moment arm estimates against a "gold standard" empirical measurement (see above) with explicit criteria for an acceptable level of precision would be valuable.

Yet like any model, simple or complex, our model is a useful starting point for a continually iterative process of improvement (enhanced by the open format of our model; *Hutchinson et al., 2015*) that progressively approximates reality. It has the advantage of

being able to estimate muscle forces, moments, and length changes that are unlikely or impossible to be measurable *in vivo*, especially simultaneously across a full stride.

## Implications for ostrich limb muscle function

Table 4 shows our classification of the major actions of ostrich pelvic limb muscles (also see *Lamas, Main & Hutchinson (2014)* for emu muscles). This approach, as opposed to the classical perspective in comparative anatomy and biomechanics, emphasizes the three-dimensional nature of avian limbs and the capacity of muscles to generate moments about many joints and degrees of freedom (see also the categorization for cat ankle muscles provided by *Young, Scott & Loeb (1993)*; also model-based assessments using isometric force-fields by *Kargo & Rome (2002)* and endpoint forces by *Burkholder & Nichols (2004)*). Standard functional anatomy papers tend to emphasize flexion/extension and only present other actions (or any actions about other joints than those deemed to be the main joints of action) in a seemingly arbitrary fashion (e.g., *Smith et al., 2007*: Table 2; cf. our Table 4), with at least the implicit assumption that non-parasagittal actions are less important. Our approach also reveals how the functions of some muscles have been misclassified before (or remain uncertain), for ostriches and perhaps for other avian species—to the degree that ostriches are representative of ancestral muscle functions, which remains to be determined. The degree of uncertainty about avian pelvic limb muscle function particularly applies to 3D actions about the hip joint; many muscles have actions (e.g., flexion/extension) that switch depending on the joint angles adopted.

Examples of somewhat surprising 3D actions of hip muscles include the M. obturatorius medialis (OM), which we find to mainly fulfill a lateral rotator and flexor action at the hip, whereas *Smith et al. (2006)*, *Smith et al. (2007)*), *Bates & Schachner (2012)*, *Bates, Benson & Falkingham (2012)*, *Bates et al. (2012)* favour an extensor action of this muscle (in addition to lateral rotation) in ostriches and other archosaurs. In the model of B.A.S., this seems to be at least partly due to a distal insertion of the OM muscle on the femur, rather than very proximally onto the trochanteric crest *Gangletal, 2004*; *Zinoviev, 2006*. Furthermore, some "antigravity muscles" act in one direction synergistically with other muscles, resisting gravity, but in other direction(s) they act antagonistically, requiring greater moments from other "antigravity" muscles. Clear examples include the PIFML and CFP muscles, which are presumably active in hip extension but incur lateral rotator and abductor moments about the hip (Table 4). These muscles probably oppose other antigravity muscles such as the ITC in medial rotation, whereas they would oppose potential hip extensors and adductors such as AMB1 (cf. moment data in *Rubenson et al., 2011*: Fig. 7), indicating the complex dynamics of hip musculature.

The analysis of *Rubenson et al.* (*2011*; Fig. 7) indicates that hip adduction, not abduction, must be resisted during stance phase in locomoting ostriches, and thus abductor muscle activity is predicted, a moment that many hip extensors create anyway. However, hip adduction capacity is far more limited—only the IC, AMB1, 2 and IFI muscles have clear actions in hip adduction. (*Smith et al. (2006)*: Table 2) assigned adductor actions to other muscles such as the flexor cruris (FC) heads, PIFML and OM whereas we find

these to be abductors. Indeed, the actions of the two heads of AMB may oppose each other (Table 4), so it would be interesting to know how they are coordinated. The ITC muscle's parts have clear roles in medial rotation, but their actions in flexion/extension vary with limb posture (see also *Gatesy, 1994*), rendering it less clear whether (or when) they play a predominant hip extensor (e.g., *Rubenson et al., 2006*) or hip flexor (e.g., *Smith et al., 2006*; *Smith et al., 2007*) role in ostriches or other birds. How any birds balance this complex interaction of long-axis and ab/adductor moments at the hip or other joints remains almost unexplored (but see *Gatesy, 1994*), yet modelling (and simulation) approaches such as ours offer one way to tease apart the complexity. *Bates & Schachner (2012)* found that *Alligator* and *Struthio* had similar hip extensor moment arms but there were large abduction and small adduction moment arms in their ostrich, along with large long-axis rotation moment arms. The functional and evolutionary implications of these differences remain unclear, and dependent on understanding force balance about the hip joint in extant archosaurs such as *Struthio*.

Complex function is not restricted to proximal muscles, however. Complex anatomy of distal limb muscles is a pernicious problem in avian locomotor biomechanics, and difficult to render realistically in biomechanical models such as ours. As the Text S1 describes, we could not model all origins (or subdivided tendons; e.g., digital flexors) of all muscle parts. Indeed, in some cases the origins are diffuse—e.g., M. gastrocnemius medialis originates mainly from the medial side of the proximal tibiotarsus, but the surrounding fascia it is attached to continue proximally past the knee joint, via the proximal patella and other structures. It is not clear if some of these distal muscles exert important moments about the knee joint (some forces may be going directly to their distal origins from the tibiotarsus), and the dynamics of the patella (not represented in our model except as a static wrapping surface) further complicates matters. Thus, it is unclear how forces are balanced across ostrich (or other avian) knees, complicating comparisons with other species (e.g., *Higham, Biewener & Wakeling, 2008*; *Andrada et al., 2013*).

*Young, Scott & Loeb (1993)* and *Johnson et al. (2008)* noted that some muscles in cat and rat hindlimbs seemed to have intrinsically stabilizing properties, shifting from flexor to extensor moment arms in a linear fashion with increasing joint flexion. Eight modelled ostrich limb muscles also show this pattern: the AMB1, AMB2, IC, ITCa, ITCp, ITM, ITCR and ISF exhibit stabilization function in flexion-extension (Figs. 9 and 10). Weaker evidence for self-stabilization is present for the OM muscle in hip ab/adduction (Fig. 14) and the four ankle flexors in flexion/extension (TCf, TCt, EDL, and FL; Fig. 18), so any self-stabilization properties must be interpreted as being largely restricted to the hip's flexion-extension function (see also Table 4). Judging from these hip muscles' paths, their long lines of action (due to the elongate pelvis and limb) running close to the hip seem to predispose them to these intrinsic stabilization properties. These patterns deserve more examination in a comparative context with other species in the future—for example, similar hip muscles show similar moment arm patterns that hint at intrinsic stabilization in various extinct dinosaurs and other archosaurian reptiles (*Hutchinson et al.,*

*2005*; *Hutchinson et al., 2008*; *Bates & Schachner, 2012*; *Bates, Benson & Falkingham, 2012*; *Bates et al., 2012*; *Maidment et al., 2013*).

## Musculoskeletal models of limb function: past, present and future

A wide variety of studies have used musculoskeletal models to reconstruct limb function in extant and extinct animals, but there remains little agreement for standards of model design, analysis and validation. The same software (SIMM) or other packages (GaitSym, Anybody, varieties of Adams, etc.) has been used to estimate limb muscle moment arms in other extant species including chimpanzees and other hominins (*O'Neill et al., 2013*; *Holowka & O'Neill, 2013*; and references therein), horses (*Brown et al., 2003a*; *Brown et al., 2003b*; *Zarucco et al., 2006*; *Harrison et al., 2010*), domestic cats (*Burkholder & Nichols, 2004*), rats (*Johnson et al., 2008*), emus (*Goetz et al., 2008*), *Alligator* and ostriches (*Bates & Schachner, 2012*), frogs (*Kargo & Rome, 2002*; *Kargo, Nelson & Rome, 2002*), cockroaches (*Full & Ahn, 1995*) and others, in addition to simpler past approaches (e.g., for small mammals, *McClearn, 1985*). It is not clear which software packages most accurately estimate muscle moment arms, but our comparisons with the data from *Bates & Schachner (2012)* suggest some advantages of our approach. Regardless, comparative biologists seeking to harness the power of musculoskeletal modelling techniques have much to learn from the considerable progress made in analyzing and validating similar models of human limb muscles (e.g., *Delp et al., 1990*; *Delp & Zajac, 1992*; *Delp et al., 1999*; *Arnold et al., 2000*; *Arnold & Delp, 2001*; *Holzbaur, Murray & Delp, 2005*; *Nikooyan et al., 2011*).

There has been a recent flourishing of musculoskeletal models of extinct taxa, especially hominins (e.g., *Australopithecus* in *Nagano et al., 2005*) and dinosaurs or other archosaurs. Theropod dinosaurs have featured prominently, including *Allosaurus*, *Struthiomimus* (*Bates & Schachner, 2012*; *Bates, Benson & Falkingham, 2012*), *Tyrannosaurus* (*Hutchinson et al., 2005*), and *Velociraptor* (*Hutchinson et al., 2008*), among other taxa (*Allen, 2010*). More recently, sauropodomorph (*Mallison, 2010a*; *Mallison, 2010b*) as well as ornithischian dinosaurs have enjoyed a new focus (*Mallison, 2010c*; *Bates et al., 2012*; *Fujiwara & Hutchinson, 2012*; *Maidment et al., 2013*). Other extinct taxa studied, for example, includethe stem-crocodile *Poposaurus* (*Bates & Schachner, 2012*) and related taxa (*Molnar, 2014*), a pterosaur (*Costa, Rocha-Barbosa & Kellner, 2013*) and the early tetrapod *Ichthyostega* (*Pierce, Clack & Hutchinson, 2012*). The detail and apparent effort put into these models varies widely, often depending on the research questions being asked. It remains debatable how detailed such models of extinct taxa need to be in order to be sufficiently accurate—are cruder models good enough for strictly comparative work? There are clear analogies with the usage of finite element models (e.g., *Rayfield, 2007*)—often, it is argued that simpler models are sufficient for simple comparative questions, whereas more complex questions such as bone failure (or, for musculoskeletal models, locomotor dynamics or performance?) require more realistic models.

Whether the focus is on limb joint ranges of motion, muscle moment arms, or dynamic simulation, there is a need for more scrutiny of models of extant taxa to establish how accurate and reliable these modelling approaches truly are, and ultimately a renewed

examination and synthesis of those findings with past studies of extinct taxa. Studies using models to estimate muscle moment arms need to consider not only this, but also how or whether those parameters actually matter for particular muscles, joints, behaviours or species (see also *Bates & Schachner, 2012*; *Bates et al., 2012*; *Maidment et al., 2013*).

To the degree that general principles of moment arm usage exist, they should clarify under what circumstances a moment arm is optimized to perform a certain function. An understanding of this link between muscle mechanics, kinematics and moment arms could link the disparate palaeobiological/comparative studies on limb joint ranges of motion and orientations (e.g., *Mallison, 2010a*; *Mallison, 2010b*; *Mallison, 2010c*; *Pierce, Clack & Hutchinson, 2012*; *Molnar, 2014*) with those of limb muscle moment arms (*Hutchinson et al., 2005*; *Hutchinson et al., 2008*; *Allen, 2010*; *Bates & Schachner, 2012*; *Bates et al., 2012*; *Fujiwara & Hutchinson, 2012*; *Maidment et al., 2013*). Such a synthesis could lead to a robust, sustainable future for comparative musculoskeletal modelling and simulation, and new insights into the evolution of musculoskeletal function and locomotor dynamics.

## ACKNOWLEDGEMENTS

We thank Peter Loan for kind and patient technical support. Thanks to Aaron Daub for his proficient assistance in the ostrich digitization and dissection. This paper benefited from discussions with F Clay Anderson, Allison Arnold, Silvia Blemker, Steve Gatesy, and Karl Bates, as well as with members of the Stanford Neuromuscular Biomechanics Laboratory and the Structure & Motion Laboratory at The Royal Veterinary College. We graciously appreciate constructive reviews by Robert Kambic and Karl Bates, and comments from Diego Sustaita, that improved the manuscript.

## APPENDIX

A few observations from our dissections revealed differences in muscle identifications from previous studies, which themselves have had much disagreement (reviewed in *Gangletal, 2004*; *Zinoviev, 2006*). However, generally our findings agree well with the excellent study by *Gangletal (2004)*. *Zinoviev (2006)* made some amendments to the latter study that our dissections independently have confirmed.

First, rather than treat "M. pectineus" as a novel muscle unrelated to M. ambiens (e.g., *Mellett, 1994*; *Gangletal, 2004*) or as a part of M. femorotibialis medialis (= FMTM or "internus"; *Zinoviev, 2006*), we refer to the muscle here as M. ambiens 1 (AMB1; for its pubic origin as in most other Reptilia; *Hutchinson, 2002*) and refer to the muscle with the derived iliac origin as M. ambiens 2 (AMB2; i.e., the dorsal head) (Table 2). We do not intend for this to be a formal nomenclatural change but it is used for convenience within this paper and to suggest that the homologies (and thus formal nomenclature) of these two muscle heads deserve reconsideration. Our nomenclature avoids confusion with the mammalian M. pectineus which has no homology with these muscles, and maintains the same nomenclature for the homologous M. ambiens (whether one or two parts) in extant Sauropsida. We deem these homologies and nomenclature marginally more parsimonious than other interpretations (reviewed by *Zinoviev, 2006*), the latest of which (favoured by *Zinoviev (2006)*) requires loss of the origin of M. ambiens from the pectineal

(preacetabular) process that is plesiomorphically present in birds (*Hutchinson, 2002*) and gain of a new head of the FMTM muscle (the "pars pectineus"). In our scenario, mainly a new head of M. ambiens (AMB2) is required; a phenomenon that is not unknown in other birds and is pervasive in Crocodylia (*Hutchinson, 2002*).

Second, we agree with *Zinoviev (2006)* that *Gangl et al (2004)* confused the two crura of "M. iliofemoralis externus" with M. iliotrochantericus caudalis, which engendered further errors in identifying other muscles. We have confirmed this from dissections of three additional ostrich cadavers, and hence the position of our M. iliotrochantericus caudalis (ITCa, ITCp in Table 2) differs from theirs (also data in *Smith et al., 2006*; *Smith et al., 2007*). We consider M. iliofemoralis externus (IFE) to be weakly differentiated from the latter muscle, also intimated by *Gangl et al* (*2004*: p. 113; and *Gadow, 1880*), but we represent it in our model as a separate head (IFE), again agreeing with *Zinoviev (2006)*. This is corroborated by the main muscle in question (ITC) having a large internal tendon, preacetabular position (between the heads of Mm. iliotibiales; IC and IL), and overlaying Mm. iliotrochanterici medius et cranialis (ITM, ITCR), which are traits diagnostic of M. iliotrochantericus caudalis, not "M. iliofemoralis externus" as in *Gangl et al (2004)*. This then explains why the latter study did not find M. iliotrochantericus medius—it identified it as M. iliotrochantericus caudalis, noting some fusion and distolateral insertion with M. iliotrochantericus cranialis, which are diagnostic of M. iliotrochantericus medius instead. Hence our M. iliotrochantericus medius (ITM) corresponds to their "M. iliotrochantericus caudalis"; but our M. iliotrochantericus cranialis (ITCR) corresponds to theirs. *Smith et al. (2006)* perpetuated the errors caused by Gangl et al.'s misidentifications, which complicates comparisons with our data (see our 'Results' section, Fig. 10). *Bates & Schachner (2012)* appear to have avoided these errors.

A third point of discordance between this study and *Gangl et al*'s (*2004*) is that we consider the latter study's "Mm. femorotibiales externus et medius" to be two parts (superficial and deep) of M. femorotibialis lateralis (FMTL; *vide Zinoviev, 2006*), because this avian muscle typically originates on the lateral surface of the femur, deep to M. iliotibialis lateralis (IL), as the former two parts do. *Gangl et al*'s (*2004*) "Mm. femorotibiales internus et accessorius" then correspond to our Mm. femorotibiales medialis et intermedius (FMTM, FMTIM), because their topological connections more closely match these muscles in birds. *Zinoviev (2006)* again gave a detailed correction that we concur with, whereas *Smith et al. (2006)*, *Smith et al. (2007)* and possibly *Bates & Schachner (2012)* used *Gangl et al*'s (*2004*) misidentifications.

Accepting these identifications renders ostrich anatomy more similar to other birds: M. iliotrochantericus caudalis remains large and originating dorsal to M. iliotrochantericus medius, which originates cranial to M. iliotrochantericus cranialis, and the insertions of these muscles retain similar proximodistal positions (the first on the lateral surface of the proximal femur, the second two sequentially distal on the craniolateral edge of the trochanteric crest; *Rowe, 1986*; *Hutchinson, 2002*). Furthermore, Mm. femorotibiales in our view then match avian anatomical positions more reasonably.

However, ostrich pelvic muscle anatomy is still derived in the large relative size of M. iliofemoralis internus (IFI; "cuppedicus" of *Rowe, 1986*; Table 2) and the two distinct heads of M. ambiens (i.e., the novel dorsal head, AMB2). Additionally, M. iliotrochantericus medius (ITM) is slightly derived in having a markedly more dorsal position (required by the dorsally shifted M. ambiens and M. iliofemoralis internus (IFI)) than in typical neornithines, and Mm. femorotibiales have strong division of the lateral head (FMTL) into two parts, but not a novel "accessorius" head. Furthermore, M. obturatorius medialis (OM) in ostriches is extraordinary in its mostly lateral origin, whereas normally in birds it is confined to the inside of the pelvis.

Similar problems with muscle identifications vex other studies of ratite limb form and function. *Patak & Baldwin*'s (*1998*) anatomical description of emu pelvic limb muscles was used to formulate *Goetz et al.*'s (*2008*) musculoskeletal model of an emu, incorporating analogous errors that deserve correction in future studies (see *Lamas, Main & Hutchinson, 2014*). The issues outlined in this appendix drive home the point that a firm grasp of avian myology, ideally based upon understanding of fundamental groups of muscles likely present in the ancestral neornithine bird and hence homologous among its living descendants, is critical for any biomechanical and comparative analyses of muscle form, function and evolution (see also *Zinoviev, 2006*).

### Funding

This work was completed as part of a postdoctoral fellowship from the National Science Foundation awarded to John R. Hutchinson in 2001, and subsequent funding from the BBSRC, grant number BB/I02204X/1, as well as the Leverhulme Trust (grant number RPG-2013-108). Additional support was provided by the Biomechanical Engineering Division at Stanford University, and from The Royal Veterinary College, Department of Comparative Biomedical Sciences. The funders had no role in study design, data collection and analysis, decision to publish, or preparation of the manuscript.

### Grant Disclosures

The following grant information was disclosed by the authors:
National Science Foundation.
BBSRC: BB/I02204X/1.
Leverhulme Trust: RPG-2013-108.

### Competing Interests

John R. Hutchinson is an Academic Editor for PeerJ.

### Author Contributions

- John R. Hutchinson and Jeffery W. Rankin conceived and designed the experiments, performed the experiments, analyzed the data, wrote the paper, prepared figures and/or tables, reviewed drafts of the paper.

- Jonas Rubenson conceived and designed the experiments, performed the experiments, contributed reagents/materials/analysis tools, wrote the paper, reviewed drafts of the paper.
- Kate H. Rosenbluth conceived and designed the experiments, performed the experiments, analyzed the data, reviewed drafts of the paper.
- Robert A. Siston conceived and designed the experiments, performed the experiments, contributed reagents/materials/analysis tools, reviewed drafts of the paper.
- Scott L. Delp conceived and designed the experiments, contributed reagents/materials/analysis tools, reviewed drafts of the paper.

### Data Deposition
The following information was supplied regarding the deposition of related data:
Figshare: http://dx.doi.org/10.6084/m9.figshare.1409443.

### Supplemental Information
Supplemental information for this article can be found online at http://dx.doi.org/10.7717/peerj.1001#supplemental-information.

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
