# Peer review of "Musculoskeletal modelling of an ostrich (Struthio camelus) pelvic limb: influence of limb orientation on muscular capacity during locomotion"

_PeerJ, doi:10.7717/peerj.1001_

## Round 0.1 · original submission · Minor Revisions

· Academic Editor

Minor Revisions

Both reviewers are generally happy with the manuscript and appreciate the data presented in the paper. There are, however, a number of points that need to be addressed to the reviewers' satisfaction, before the paper could be accepted for publication. Since we already know about possible issues with the data from the literature that you have used for comparison with your own data, it is important that you stay in close contact with the editorial office to make sure the protocols of the journal are properly followed.

·

Basic reporting

There are a few minor cases where features on plots are not explained in the caption, or conflict with captions/tables; these have been noted in the uploaded annotated pdf. Additionally, the supplemental movie should be slowed down. Savvy viewers will step through the movie, but those who don't won't get much from it currently.

Experimental design

No comments

Validity of the findings

No comments

Additional comments

This paper presents a wealth of new data for those of us who are interested in 3-D avian locomotion. The paper's greatest strengths are in presenting the results in the context of earlier work while describing limitations of the data and challenges of the approach. I do not believe the paper requires substantial revision. Most of my comments relate to small oversights or changes that I believe will make things easier on readers. There are a few places where the methods should be more clear. All of these are covered in the annotated pdf I have attached.

·

Basic reporting

No comments.

Experimental design

No comments.

Validity of the findings

No Comments

Additional comments

This is an excellent paper and makes an important contribution to a number of topics in vertebrate anatomy, biomechanics and evolution. I applaud the authors for presenting this study in full detail so that the various methodological issues they raise can be fully and explicitly discussed. In particular they do a great job of objectively comparing the 3 studies, and this comparison raises important issues for studying animal locomotion in general. I have only minor comments, which the authors may or may not consider making in a re-submission.

Page 3 ln 92: Typo. delete random “o” in the middle of the sentence (“…we investigate how pelvic muscle functions o relate to limb orientation…”).

Page 3 ln 111 (and throughout): there are two Brassey et al. 2013’s. Unless PeerJ referencing style precludes it, it would be nice to label them ‘a’ and ‘b’ so the reader knows which is being referred to.

Page 12 lines 572-575: Surprisingly, ankle extensors reveal more variation: our FDL’s ankle extensor moment arm is almost twice as large of that in the S.E.A. and B.A.S. data, showing little change with ankle posture, whereas the B.A.S. dataset exhibited a decreased moment arm with flexion." This is the restriction of modelling muscles with only via points OR a single wrapping surface that existed in GaitSym when this model was made (more flexibility now exists). I had to just via points rather than wrapping surface/cylinder at the ankle, hence the moment arm decreases as the joint is flexed (it obviously couldn't if a cylinder was used, which is what I would do now).

Page 14, final paragraph, discussion of fibre length-moment arm tuning. Is it worth mentioning the small analysis of Sellers et al. (2013, PLoS ONE: http://www.plosone.org/article/info%3Adoi%2F10.1371%2Fjournal.pone.0078733) regarding fibre length: length changes?

Page 15 lines 712-714: Whilst I would normally applaud healthy cynicism/admitting what we don’t know, I wonder if the authors might re-consider the exact wording here. I agree completely with the general sentiment (i.e. we don’t really understand everything about living animals, hence studying extinct animals is VERY difficult), but I wonder if the current wording here gives too much ammunition to those conceptually opposed to applying biomechanical approaches to fossils? i.e. better to explicitly say it remains the most robust and honest/open approach to testing specific hypotheses (versus more pragmatic/qualitative approaches), but we’re a long way doing it well it.

Page 16, paragraph 2 (beginning “Contrastringly, the “M. femorotibialis medialis”…). I found this very difficult to read, and Im still not sure I completely understand it (i.e. which data is consistent with which, which is likely wrong etc.). Could this be simplified/made a little clearer?

Page 21, paragraph 1: Specifically, the comment “It is not clear which software packages most accurately estimate moment arms….” This made me think about the various sources of “error” and how/why/to what extent they may be introduced. For me, accuracy is less likely to be down to software than it is the researchers and the skill, time and effort put into the model. I accept that different software packages have different functionalities/options for modeling muscles (e.g. back in 2010 when I made the Bates & Schachner model, GaitSym lacked the facilities to wrap muscles with multiple cylinders and/or cylinders AND via points; it was one wrapping surface or via points, not both), it should still be possible to for two independent teams to converge on qualitatively “identical” and quantitatively similar results for moment arms, as has occurred here for the vast majority of flexion-extension moment arms. I believe that time and effort will ultimately be shown to be responsible for most of the quantitative differences between the models. Perhaps the authors can’t say this so openly in print, but their model is almost certainly more accurate/better than the Bates & Schachner model because far greater resources were put into building it (particularly the digitization during dissection and also the estimation of joint axes/centers of rotation). Also, something not discussed at present, is specifically how the motivation of the study influences how a model is built: In the present study, the ostrich model is constructed as an ostrich, i.e. using all the data available for a living animal (motivation = lets invest as much resources as possible into building the best ostrich model possible). This differs slightly from the Bates and Schachner model, which was built more as if it were an extinct animal, and was generally much cruder (motivation = lets build the best ostrich model we can in same way we build a model for an extinct animal). The authors may disagree, but I think that the present results provide preliminary support that the crude approach is probably good enough (at least for most muscles) for the kind of higher-level comparisons/qualitative conclusions that Bates and Schachner sought to make. I guess if it we me I refocus the discussion a little to downplay software related error and make more investigator time/effort/resources and motivation issues, but this is obviously up to the authors to decide.

---

## Round 0.2 · accepted · Accept

· Academic Editor

Accept

Thank you for the revisions and addressing the reviewers' comments. All comments have been satisfactorily addressed.